# BYZANTINE CLUSTER-SENDING IN EXPECTED CONSTANT COST AND CONSTANT TIME

JELLE HELLINGS
*Department of Computing and Software*
*McMaster University*
*jhellings@mcmaster.ca*

MOHAMMAD SADOGHI
*Exploratory Systems Lab*
*Department of Computer Science*
*University of California, Davis*
*msadoghi@ucdavis.edu*

## Abstract

Traditional resilient systems operate on fully-replicated fault-tolerant clusters, which limits their scalability and performance. One way to make the step towards resilient high-performance systems that can deal with huge workloads is by enabling independent fault-tolerant clusters to efficiently communicate and cooperate with each other, as this also enables the usage of high-performance techniques such as sharding. Recently, such inter-cluster communication was formalized as the *Byzantine cluster-sending problem*. Unfortunately, existing worst-case optimal protocols for cluster-sending all have *linear complexity* in the size of the clusters involved.

In this paper, we propose *probabilistic cluster-sending techniques* as a solution for the cluster-sending problem with only an *expected constant message complexity*, this independent of the size of the clusters involved and this even in the presence of highly unreliable communication. Depending on the robustness of the clusters involved, our techniques require only *two-to-four* message round-trips (without communication failures). Furthermore, our protocols can support worst-case linear communication between clusters. Finally, we have put our techniques to the test in an in-depth experimental evaluation that further underlines the exceptional low expected costs of our techniques in comparison with other protocols. As such, our work provides a strong foundation for the further development of resilient high-performance systems.

## 1 Introduction

The promises of *resilient data processing*, as provided by private and public blockchains [15, 23, 29, 30], has renewed interest in traditional consensus-based Byzantine fault-tolerant resilient systems [5, 6, 26]. Unfortunately, blockchains and other consensus-based systems typically rely on fully-replicated designs, which limits their scalability and performance. Consequently, these systems cannot deal with the ever-growing requirements in data processing [32, 33].

One way to improve on these limitations is by building complex system designs that consist of *independently-operating* resilient clusters that can cooperate to provide certain services. To illustrate this, one can consider a sharded resilient design. In a traditional resilient systems, resilience is provided by a fully-replicated consensus-based Byzantine fault-tolerant cluster in which all replicas hold all data and process all requests. This traditional design has only limited performance, even with the best consensus protocols, and lacks scalability. To improve on the design of traditional systems, one can employ the *sharded* design of Figure 1. In this sharded design, each cluster only holds part of the data. Consequently, each cluster only needs to process requests that affect data they hold. In this way, this sharded design improves performance by enabling *parallel processing* of requests by different clusters, while also improving storage scalability. To support *arbitrary general-purpose workloads* that can affect data in several clusters in such a sharded design, the clusters need to be able to *coordinate their operations*, however [1, 7, 16, 19, 21].[1]

Central to such complex system designs is the ability to reliably and efficiently communicate between independently-operating resilient clusters. Recently, this problem of communication *between* Byzantine fault-tolerant clusters has been formalized as the *cluster-sending problem* [18, 20]. We believe that efficient solutions to this problem have a central role towards bridging *resilient* and *high-performance* data processing.

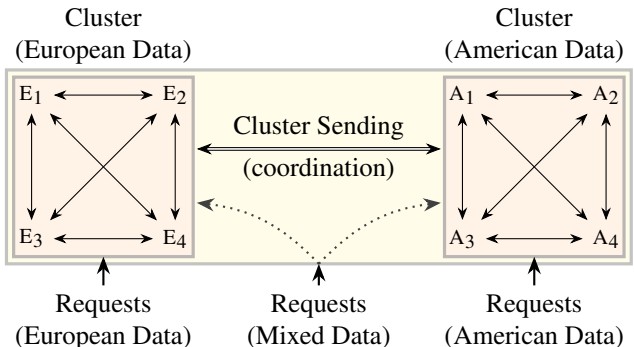

Cluster (European Data)    Cluster (American Data)

Cluster Sending (coordination)

Requests (European Data)    Requests (Mixed Data)    Requests (American Data)

Figure 1: A *sharded* design in which each resilient cluster of four replicas holds only a part of the data. Local decisions within a cluster are made via *consensus* (⟷), whereas multi-shard coordination to process multi-shard transactions requires *cluster-sending* (⟺). This illustration is based on Hellings et al. [20, Figure 1].

Although the cluster-sending problem has received some attention (e.g., as part of the design of AHL [7], BYSHARD [19,

---

[1]Strict ordering as provided by consensus is necessary to support arbitrary general-purpose workloads. There are classes of operations for which strict consensus-based ordering of (sharding) steps is unnecessary, however. Examples include balance changes and, more generally, operations on CRDTs [34].

21], GEOBFT [16], and CHAINSPACE [1]), and cluster-sending protocols that solve the cluster-sending problem with worst-case optimal complexity are known [18, 20], we believe there is still much room for improvement.

In this paper, we introduce a new solution to the cluster-sending problem: we introduce cluster-sending protocols that use *probabilistic cluster-sending* techniques and are able to provide low *expected-case* message complexity (at the cost of higher communication latencies, a good trade-off in systems where inter-cluster network bandwidth is limited). To simplify presentation, we first show how probabilistic cluster-sending works when communication is reliable and synchronous. Then, we generalize these synchronous solutions to practical environments in which communication can be unreliable and asynchronous. Our main contributions are as follows:

1. First, in Section 3, we introduce the cluster-sending step CS-STEP that attempts to send a value from a replica in the sending cluster to a replica in the receiving cluster in a verifiable manner and with a constant amount of inter-cluster communication. This step is guaranteed to perform cluster-sending if communication is reliable and the step is performed by non-faulty replicas.

2. Then, in Section 4, we illustrate the working of a *basic probabilistic cluster-sending protocol* by introducing the *Synchronous Probabilistic Cluster-Sending protocol* PCS. PCS uses CS-STEP with randomly selected sending and receiving replicas to provide cluster-sending in *expected constant* steps. In addition, we show how *pruned*-PCS (PPCS), a fine-tuned variant of PCS, can guarantee cluster-sending in *expected constant* steps while also guaranteeing termination.

3. Next, in Section 5, we propose the *Synchronous Probabilistic Linear Cluster-Sending protocol* PLCS. PLCS not only guarantees cluster-sending in *expected constant* steps, but also guarantees a *worst-case optimal* linear upper-bound on communication. To achieve this worst-case optimal upper-bound, we introuce a specialized randomized scheme via which PLCS selects replicas. To prove the complexity bounds of PLCS, we provide an in-depth analysis of the expected behavior of the randomized scheme we introduce.

4. Next, in Section 6, we generalize PCS, PPCS, and PLCS to practical environments in which communication can be *unreliable and asynchronous*.

5. Finally, in Section 7, we evaluate the behavior of the proposed probabilistic cluster-sending protocols via an in-depth evaluation. In this evaluation, we show that probabilistic cluster-sending protocols has exceptionally low communication costs in comparison with existing cluster-sending protocols, this even in the presence of communication failures.

A summary of our findings in comparison with existing techniques can be found in Figure 2. In Section 2, we introduce the necessary terminology and notation, in Section 8, we compare with related work, and in Section 9, we conclude on our findings.

## 2 The Cluster-Sending Problem

Before we present our probabilistic cluster-sending techniques, we first introduce all necessary terminology and notation. The formal model we use is based on the formalization of the cluster-sending problem provided by Hellings et al. [18, 20]. If $S$ is a set of replicas, then $\mathsf{f}(S) \subseteq S$ denotes the *faulty replicas* in $S$, whereas $\mathsf{nf}(S) = S \setminus \mathsf{f}(S)$ denotes the *non-faulty replicas* in $S$. We write $\mathbf{n}_S = |S|$, $\mathbf{f}_S = |\mathsf{f}(S)|$, and $\mathbf{nf}_S = |\mathsf{nf}(S)| = \mathbf{n}_S - \mathbf{f}_S$ to denote the number of replicas, faulty replicas, and non-faulty replicas in $S$, respectively. A *cluster* $\mathcal{C}$ is a finite set of replicas. We consider clusters with *Byzantine replicas* that behave in arbitrary manners. In specific, if $\mathcal{C}$ is a cluster, then any malicious adversary can control the replicas in $\mathsf{f}(\mathcal{C})$ at any time, but adversaries cannot bring non-faulty replicas under their control.

**Definition 2.1.** Let $\mathcal{C}_1, \mathcal{C}_2$ be disjoint clusters. The *cluster-sending problem* is the problem of reliably sending a value $v$ from $\mathcal{C}_1$ to $\mathcal{C}_2$ in the presence of failures and malicious behaviour: non-faulty replicas in $\mathcal{C}_2$ can only receive $v$ if the non-faulty replicas in $\mathcal{C}_1$ reached agreement on sending $v$, and the non-faulty replicas in $\mathcal{C}_1$ receive a confirmation only if the non-faulty replicas in $\mathcal{C}_2$ received $v$.

We formalize the *cluster-sending problem* of sending a value $v$ from $\mathcal{C}_1$ to $\mathcal{C}_2$ as a progression of replica states as follows:

1. Cluster-sending is initiated when the non-faulty replicas in $\mathcal{C}_1$ reach the state AGREE($v$) indicating that the replicas in $\mathcal{C}_1$ have internally decided upon sending $v$ to $\mathcal{C}_2$.

2. Only after all non-faulty replicas in $\mathcal{C}_1$ reach the state AGREE($v$), the non-faulty replicas in $\mathcal{C}_2$ can reach the state RECEIVE($v$).

3. Only after all non-faulty replicas in $\mathcal{C}_2$ reach the state RECEIVE($v$), the replicas in $\mathcal{C}_1$ can reach the state CONFIRM($v$).

Cluster-sending is succesfull if all non-faulty replicas in $\mathcal{C}_1$ reach the state CONFIRM($v$).

We note that the value $v$ sent by cluster $\mathcal{C}_1$ is a parameter of the cluster-sending problem. Typically, the value $v$ originates from the resilient applications that are running on the clusters

Figure 2: A comparison of *cluster-sending protocols* that send a value from cluster $C_1$ with $\mathbf{n}_{C_1}$ replicas, of which $\mathbf{f}_{C_1}$ are faulty, to cluster $C_2$ with $\mathbf{n}_{C_2}$ replicas, of which $\mathbf{f}_{C_2}$ are faulty. For each protocol $P$, *Protocol* specifies its name; *Robustness* specifies the conditions $P$ puts on the clusters; *Message Steps* specifies the number of messages exchanges $P$ performs; *Optimal* specifies whether $P$ is worst-case optimal; and *Unreliable* specifies whether $P$ can deal with unreliable communication.

| | Protocol | Robustness[a] | Message Steps | | Optimal | Unreliable |
|---|---|---|---|---|---|---|
| | | | (expected-case) | (worst-case) | | |
| | PBS-CS [18, 20] | $\min(\mathbf{n}_{C_1}, \mathbf{n}_{C_2}) > \mathbf{f}_{C_1} + \mathbf{f}_{C_2}$ | $\mathbf{f}_{C_1} + \mathbf{f}_{C_2} + 1$ | | ✔ | ✘ |
| | PBS-CS [18, 20] | $\mathbf{n}_{C_1} > 3\mathbf{f}_{C_1}, \mathbf{n}_{C_2} > 3\mathbf{f}_{C_2}$ | $\max(\mathbf{n}_{C_1}, \mathbf{n}_{C_2})$ | | ✔ | ✘ |
| | GEOBFT [16] | $\mathbf{n}_{C_1} = \mathbf{n}_{C_2} > 3\max(\mathbf{f}_{C_1}, \mathbf{f}_{C_2})$ | $\mathbf{f}_{C_2} + 1$[b] | $\Omega(\mathbf{f}_{C_1}\mathbf{n}_{C_2})$ | ✘ | ✔ |
| | CHAINSPACE [1] | $\mathbf{n}_{C_1} > 3\mathbf{f}_{C_1}, \mathbf{n}_{C_2} > 3\mathbf{f}_{C_2}$ | $\mathbf{n}_{C_1}\mathbf{n}_{C_2}$ | | ✘ | ✘ |
| This Paper | PPCS | $\mathbf{n}_{C_1} > 2\mathbf{f}_{C_1}, \mathbf{n}_{C_2} > 2\mathbf{f}_{C_2}$ | 4 | $(\mathbf{f}_{C_1} + 1)(\mathbf{f}_{C_2} + 1)$ | ✘ | ✔ |
| | PPCS | $\mathbf{n}_{C_1} > 3\mathbf{f}_{C_1}, \mathbf{n}_{C_2} > 3\mathbf{f}_{C_2}$ | $\frac{9}{4} (= 2\frac{1}{4})$ | $(\mathbf{f}_{C_1} + 1)(\mathbf{f}_{C_2} + 1)$ | ✘ | ✔ |
| | PLCS | $\min(\mathbf{n}_{C_1}, \mathbf{n}_{C_2}) > \mathbf{f}_{C_1} + \mathbf{f}_{C_2}$ | 4 | $\mathbf{f}_{C_1} + \mathbf{f}_{C_2} + 1$ | ✔ | ✔ |
| | PLCS | $\min(\mathbf{n}_{C_1}, \mathbf{n}_{C_2}) > 2(\mathbf{f}_{C_1} + \mathbf{f}_{C_2})$ | $\frac{9}{4} (= 2\frac{1}{4})$ | $\mathbf{f}_{C_1} + \mathbf{f}_{C_2} + 1$ | ✔ | ✔ |
| | PLCS | $\mathbf{n}_{C_1} > 3\mathbf{f}_{C_1}, \mathbf{n}_{C_2} > 3\mathbf{f}_{C_2}$ | 3 | $\max(\mathbf{n}_{C_1}, \mathbf{n}_{C_2})$ | ✔ | ✔ |

[a]Protocols that have different message step complexities depending on the robustness assumptions have been included for each of the robustness assumptions.
[b]Complexity when the coordinating primary in $C_1$ is non-faulty and communication is reliable.

$C_1$, e.g., due to the application requiring communication with other clusters and agree upon initiation such communication via an internal consensus step.

We assume that there is no limitation on local communication within a cluster, while global communication between clusters is costly. This model is supported by practice, where communication between wide-area deployments of clusters is up-to-two orders of magnitude more expensive than communication within a cluster [7, 16].

We assume that each cluster can make *local decisions* among all non-faulty replicas, e.g., via a *consensus protocol* such as PBFT (when Byzantine fault tolerance is required) or PAXOS [6, 26] (when crash-fault tolerance suffices). Furthermore, we assume that the replicas in each cluster can certify such local decisions via a *signature scheme*. For example, a cluster $C$ can certify a consensus decision on some message $m$ by collecting a set of signatures for $m$ of $\mathbf{f}_C + 1$ replicas in $C$, guaranteeing one such signature is from a non-faulty replica (which would only signs values on which consensus is reached). We write $\langle m \rangle_C$ to denote a message $m$ certified by $C$. To minimize the size of certified messages, one can utilize a threshold signature scheme [35]. To enable decision making and message certification, we assume, for every cluster $C$, $\mathbf{n}_C > 2\mathbf{f}_C$, a minimal requirement [9, 27]. Lastly, we assume that there is a common source of randomness for all non-faulty replicas in the sending cluster $C_1$, e.g., via a distributed fault-tolerant random coin [3, 4].

## 3 The Cluster-Sending Step

As the first step toward a probabilistic cluster-sending protocol, we introduce the *cluster-sending step* that takes as input a value $v$ and a choice of replicas $R_1 \in C_1$ and $R_2 \in C_2$, and tries to perform cluster-sending via communication between

the pair of replicas $(R_1, R_2)$. The design of the cluster-sending step will ensure that cluster-sending succeeds if both $R_1$ and $R_2$ are non-faulty, but can detectably fail otherwise. In the following sections, we introduce cluster-sending protocols that *instantiate this cluster-sending step* via one-or-more pairs of replicas $(R_1, R_2)$ to ensure that eventually a cluster-sending step succeeds.

If communication is reliable and one knows non-faulty replicas $R_1 \in \mathsf{nf}(C_1)$ and $R_2 \in \mathsf{nf}(C_2)$, then cluster-sending a value $v$ from $C_1$ to $C_2$ can be done via a straightforward *cluster-sending step*: one can simply instruct $R_1$ to send $v$ to $R_2$. When $R_2$ receives $v$, it can disperse $v$ locally in $C_2$. Unfortunately, we do not know which replicas are faulty and which are non-faulty. Furthermore, it is practically impossible to reliably determine which replicas are non-faulty, as non-faulty replicas can appear faulty due to unreliable communication, while faulty replicas can appear well-behaved to most replicas, while interfering with the operations of only some non-faulty replicas.

To deal with faulty replicas when utilizing the above *cluster-sending step*, one needs a sufficient safeguards to detect *failure* of $R_1$, of $R_2$, or of the communication between them. To do so, we add receive and confirmation phases to the sketched cluster-sending step. During the *receive phase*, the receiving replica $R_2$ must construct a proof $P$ that it received and dispersed $v$ locally in $C_2$ and then send this proof back to $R_1$. Finally, during the *confirmation phase*, $R_1$ can utilize $P$ to prove to all other replicas in $C_1$ that the cluster-sending step was successful. The pseudo-code of this *cluster-sending step protocol* CS-STEP can be found in Figure 3. We have the following:

**Proposition 3.1.** *Let* $C_1, C_2$ *be disjoint clusters with* $R_1 \in C_1$ *and* $R_2 \in C_2$. *If* $C_1$ *satisfies the pre-conditions of* CS-STEP($R_1$, $R_2, v$), *then execution of* CS-STEP($R_1, R_2, v$) *satisfies the post-*

**Protocol** CS-STEP($R_1$, $R_2$, $v$), with $R_1 \in \mathcal{C}_1$ and $R_2 \in \mathcal{C}_2$:

---

**Pre:** Each replica in nf($\mathcal{C}_1$) decided AGREE($v$) (and can construct $\langle \mathtt{send} : v,\ \mathcal{C}_2 \rangle_{\mathcal{C}_1}$).

**Post:** **(i)** If communication is reliable, $R_1 \in$ nf($\mathcal{C}_1$), and $R_2 \in$ nf($\mathcal{C}_2$), then $R_2$ decides RECEIVE($v$). **(ii)** If a replica in nf($\mathcal{C}_2$) decides RECEIVE($w$) for any value $w$, then that must have been preceded by the non-faulty replicas in $\mathcal{C}_1$ deciding AGREE($w$). **(iii)** If a non-faulty replica in nf($\mathcal{C}_2$) decides RECEIVE($v$), then all replicas in nf($\mathcal{C}_2$) will decides RECEIVE($v$) and all replicas in nf($\mathcal{C}_1$) will decide CONFIRM($v$) (whenever communication becomes reliable).

**The _cluster-sending step_ for $R_1$ and $R_2$:**

1:  Instruct $R_1$ to send $\langle \mathtt{send} : v,\ \mathcal{C}_2 \rangle_{\mathcal{C}_1}$ to $R_2$.

**The _receive role_ for $\mathcal{C}_2$:**

2:  **event** $R_2 \in$ nf($\mathcal{C}_2$) receives message $m := \langle \mathtt{send} : v,\ \mathcal{C}_2 \rangle_{\mathcal{C}_1}$ from $R_1 \in \mathcal{C}_1$ **do**
3:      **if** $R_2$ does not have consensus on $m$ **then**
4:          Use _local consensus_ on $m$ and construct $\langle \mathtt{proof} : m \rangle_{\mathcal{C}_2}$.
5:          {_Each replica in_ nf($\mathcal{C}_2$) _decides_ RECEIVE($v$).}
6:      Send $\langle \mathtt{proof} : m \rangle_{\mathcal{C}_2}$ to $R_1$.

**The _confirmation role_ for $\mathcal{C}_1$:**

7:  **event** $R_1 \in$ nf($\mathcal{C}_1$) receives message $m_p := \langle \mathtt{proof} : m \rangle_{\mathcal{C}_2}$ with $m := \langle \mathtt{send} : v,\ \mathcal{C}_2 \rangle_{\mathcal{C}_1}$ from $R_2 \in \mathcal{C}_2$ **do**
8:      **if** $R_1$ does not have consensus on $m_p$ **then**
9:          Use _local consensus_ on $m_p$.
10:         {_Each replica in_ nf($\mathcal{C}_1$) _decides_ CONFIRM($v$).}

---

Figure 3: The cluster-sending step protocol CS-STEP($R_1$, $R_2$, $v$). In this protocol, $R_1$ tries to send $v$ to $R_2$, which will succeed if both $R_1$ and $R_2$ are non-faulty.

_conditions and will exchange at most two messages between $\mathcal{C}_1$ and $\mathcal{C}_2$._

_Proof._ We prove the three post-conditions separately. **(i)** We assume that communication is reliable, $R_1 \in$ nf($\mathcal{C}_1$), and $R_2 \in$ nf($\mathcal{C}_2$). Hence, $R_1$ sends message $m := \langle \mathtt{send} : v,\ \mathcal{C}_2 \rangle_{\mathcal{C}_1}$ to $R_2$ (Line 1 of Figure 3). In the receive phase (Lines 2–6 of Figure 3), replica $R_2$ _receives_ message $m$ from $R_1$. Replica $R_2$ uses local consensus on $m$ to replicate $m$ among all replicas $\mathcal{C}_2$ and, along the way, to constructs a _proof of receipt $m_p := \langle \mathtt{proof} : m \rangle_{\mathcal{C}_2}$._ As all replicas in nf($\mathcal{C}_2$) participate in this local consensus, all replicas in nf($\mathcal{C}_2$) will decide RECEIVE on $v$ from $\mathcal{C}_1$. Finally, the proof $m_p$ is returned to $R_1$. In the confirmation phase (Lines 7–10 of Figure 3), replica $R_1$ receives the proof of receipt $m_p$. Next, $R_1$ uses local consensus on $m_p$ to replicate $m_p$ among all replicas in nf($\mathcal{C}_1$), after which all replicas in nf($\mathcal{C}_1$) decide CONFIRM on sending $v$ to $\mathcal{C}_2$

**(ii)** A replica in nf($\mathcal{C}_2$) only decides RECEIVE on $v$ after consensus is reached on a message $m := \langle \mathtt{send} : v,\ \mathcal{C}_2 \rangle_{\mathcal{C}_1}$ (Line 5 of Figure 3). This message $m$ not only contains the value $v$, but also the identity of the recipient cluster $\mathcal{C}_2$. Due

to the usage of certificates and the pre-condition, the message $m$ cannot be created without the replicas in nf($\mathcal{C}_1$) deciding AGREE on sending $v$ to $\mathcal{C}_2$.

**(iii)** A replica in nf($\mathcal{C}_1$) only decides CONFIRM on $v$ after consensus is reached on a _proof of receipt_ message $m_p := \langle \mathtt{proof} : m \rangle_{\mathcal{C}_2}$ (Line 10 of Figure 3). This consensus step will complete for all replicas in $\mathcal{C}_1$ whenever communication becomes reliable. Hence, all replicas in nf($\mathcal{C}_1$) will eventually decide CONFIRM on $v$. Due to the usage of certificates, the message $m_p$ cannot be created without cooperation of the replicas in nf($\mathcal{C}_2$). The replicas in nf($\mathcal{C}_2$) only cooperate in constructing $m_p$ as part of the consensus step of Line 4 of Figure 3. Upon completion of this consensus step, all replicas in nf($\mathcal{C}_2$) will decide RECEIVE on $v$. $\square$

In the following sections, we show how to use the cluster-sending step in the construction of cluster-sending protocols. In Section 4, we introduce synchronous protocols that provide _expected constant message complexity_. Then, in Section 5, we introduce synchronous protocols that additionally provide _worst-case linear message complexity_, which is optimal. Finally, in Section 6, we show how to extend the presented techniques to asynchronous communication.

## 4  Probabilistic Cluster-Sending with Random Replica Selection

In the previous section, we introduced CS-STEP, the cluster-sending step protocol that succeeds whenever the participating replicas are non-faulty and communication is reliable. In this section, we introduce a basic probabilistic cluster-sending protocol that utilizes CS-STEP to perform cluster-sending with expected constant costs.

Using CS-STEP, we build a three-step protocol that cluster-sends a value $v$ from $\mathcal{C}_1$ to $\mathcal{C}_2$:

1. First, the replicas in nf($\mathcal{C}_1$) reach agreement and decide AGREE on sending $v$ to $\mathcal{C}_2$.

2. Then, the replicas in nf($\mathcal{C}_1$) perform a _probabilistic cluster-sending step_ by electing replicas $R_1 \in \mathcal{C}_1$ and $R_2 \in \mathcal{C}_2$ fully at random, after which CS-STEP($R_1$, $R_2$, $v$) is executed.

3. Finally, each replica in nf($\mathcal{C}_1$) waits for the completion of CS-STEP($R_1$, $R_2$, $v$). If the waiting replicas decided CONFIRM on $v$ during this wait, then cluster-sending is successful. Otherwise, we repeat the previous step.

To simplify presentation, we first present the above protocol assuming _synchronous_ inter-cluster communication: in this sectiom, we assume that messages sent by non-faulty replicas will be delivered within some known bounded delay. _Synchronous_ systems can be modeled by _pulses_ [10,11]:

**Protocol** $\text{PCS}(\mathcal{C}_1, \mathcal{C}_2, v)$:

---

1: Use *local consensus* on $v$ and construct $\langle \texttt{send} : v, \ \mathcal{C}_2 \rangle_{\mathcal{C}_1}$.
2: {*Each replica in* $\text{nf}(\mathcal{C}_1)$ *decides* AGREE *on* $v$.}
3: **repeat**
4:    Choose replicas $(R_1, R_2) \in \mathcal{C}_1 \times \mathcal{C}_2$, fully at random.
5:    CS-STEP$(R_1, R_2, v)$
6:    Wait *three* global pulses.
7: **until** $\mathcal{C}_1$ reaches consensus on $\langle \texttt{proof} : \langle \texttt{send} : v, \ \mathcal{C}_2 \rangle_{\mathcal{C}_1} \rangle_{\mathcal{C}_2}$.

---

Figure 4: The Synchronous Probabilistic Cluster-Sending protocol $\text{PCS}(\mathcal{C}_1, \mathcal{C}_2, v)$ that cluster-sends a value $v$ from $\mathcal{C}_1$ to $\mathcal{C}_2$.

**Definition 4.1.** A system is *synchronous* if all inter-cluster communication happens in *pulses* such that every message sent in a pulse will be received in the same pulse.

We refer to Section 6 on how to generalize the results of this section to practical environments with asynchronous and unreliable communication.

The pseudo-code of the resultant *Synchronous Probabilistic Cluster-Sending protocol* PCS can be found in Figure 4. Next, we prove that PCS is correct and has expected-case constant message complexity:

**Theorem 4.2.** *Let* $\mathcal{C}_1, \mathcal{C}_2$ *be disjoint clusters. If communication is synchronous, then* $\text{PCS}(\mathcal{C}_1, \mathcal{C}_2, v)$ *results in cluster-sending* $v$ *from* $\mathcal{C}_1$ *to* $\mathcal{C}_2$. *The execution performs two local consensus steps in* $\mathcal{C}_1$, *one local consensus step in* $\mathcal{C}_2$, *and is expected to make* $(\mathbf{n}_{\mathcal{C}_1} \mathbf{n}_{\mathcal{C}_2})/(\mathbf{nf}_{\mathcal{C}_1} \mathbf{nf}_{\mathcal{C}_2})$ *cluster-sending steps.*[2]

*Proof.* Due to Lines 1–2 of Figure 4, $\text{PCS}(\mathcal{C}_1, \mathcal{C}_2, v)$ establishes the pre-conditions for any execution of CS-STEP$(R_1, R_2, v)$ with $R_1 \in \mathcal{C}_1$ and $R_2 \in \mathcal{C}_2$. Using the correctness of CS-STEP (Proposition 3.1), we conclude that $\text{PCS}(\mathcal{C}_1, \mathcal{C}_2, v)$ results in cluster-sending $v$ from $\mathcal{C}_1$ to $\mathcal{C}_2$ whenever the replicas $(R_1, R_2) \in \mathcal{C}_1 \times \mathcal{C}_2$ chosen at Line 4 of Figure 4 are non-faulty. As the replicas $(R_1, R_2) \in \mathcal{C}_1 \times \mathcal{C}_2$ are chosen fully at random, we have probability $p_i = \mathbf{nf}_{\mathcal{C}_i}/\mathbf{n}_{\mathcal{C}_i}$, $i \in \{1, 2\}$, of choosing $R_i \in \text{nf}(\mathcal{C}_i)$. The probabilities $p_1$ and $p_2$ are independent of each other. Consequently, the probability of choosing $(R_1, R_2) \in \text{nf}(\mathcal{C}_1) \times \text{nf}(\mathcal{C}_2)$ is $p = p_1 p_2 = (\mathbf{nf}_{\mathcal{C}_1} \mathbf{nf}_{\mathcal{C}_2})/(\mathbf{n}_{\mathcal{C}_1} \mathbf{n}_{\mathcal{C}_2})$. As such, each iteration of the loop at Line 3 of Figure 4 can be modeled as an independent *Bernoulli trial* with probability of success $p$, and the expected number of iterations of the loop is $p^{-1} = (\mathbf{n}_{\mathcal{C}_1} \mathbf{n}_{\mathcal{C}_2})/(\mathbf{nf}_{\mathcal{C}_1} \mathbf{nf}_{\mathcal{C}_2})$.

Finally, we prove that each local consensus step needs to be performed only once. To do so, we consider the local consensus steps triggered by the loop at Line 3 of Figure 4. These are the local consensus steps at Lines 4 and 9 of Figure 3. The local consensus step at Line 4 can be initiated

by a faulty replica $R_2$. After this single local consensus step reaches consensus on message $m := \langle \texttt{send} : v, \ \mathcal{C}_2 \rangle_{\mathcal{C}_1}$, each replica in $\text{nf}(\mathcal{C}_2)$ reaches consensus on $m$, decides RECEIVE on $v$, and can construct $m_p := \langle \texttt{proof} : m \rangle_{\mathcal{C}_2}$, this independent of the behavior of $R_2$. Hence, a single local consensus step for $m$ in $\mathcal{C}_2$ suffices, and no replica in $\text{nf}(\mathcal{C}_2)$ will participate in future consensus steps for $m$. An analogous argument proves that a single local consensus step for $m_p$ in $\mathcal{C}_1$, performed at Line 9 of Figure 3, suffices.  $\square$

*Remark* 4.3. Although Theorem 4.2 indicates local consensus steps in clusters $\mathcal{C}_1$ and $\mathcal{C}_2$, these local consensus steps typically come for *free* as part of the protocol that uses cluster-sending as a building block. To see this, we consider a multi-shard transaction $\tau$ processed by clusters $\mathcal{C}_1$ and $\mathcal{C}_2$.

The decision of cluster $\mathcal{C}_1$ to send a value $v$ to cluster $\mathcal{C}_2$ is a consequence of the execution of $\tau$ in $\mathcal{C}_1$. Before the replicas in $\mathcal{C}_1$ execute $\tau$, they need to reach consensus on the order in which $\tau$ is executed in $\mathcal{C}_1$. As part of this consensus step, the replicas in $\mathcal{C}_1$ can also construct $\langle \texttt{send} : v, \ \mathcal{C}_2 \rangle_{\mathcal{C}_1}$ without additional consensus steps. Hence, no consensus step is necessary in $\mathcal{C}_1$ to send value $v$. Likewise, if value $v$ is received by replicas in $\mathcal{C}_2$ as part of some multi-shard transaction execution protocol, then the replicas in $\mathcal{C}_2$ need to perform their portion of the necessary transaction execution steps to execute $\tau$ as a *consequence* of receiving $v$. To do so, the replicas in $\mathcal{C}_2$ need to reach consensus on the order in which these transaction execution steps are performed. As part of this consensus step, the replicas in $\mathcal{C}_2$ can also constructing a proof of receipt for $v$.

In typical fault-tolerant clusters, more than half of the replicas are non-faulty (e.g., in synchronous systems with Byzantine failures that use digital signatures, or in systems that only deal with crashes) or more than two-third of the replicas are non-faulty (e.g., asynchronous systems). In these systems, PCS is expected to only performs a few cluster-sending steps:

**Corollary 4.4.** *Let* $\mathcal{C}_1, \mathcal{C}_2$ *be disjoint clusters. If communication is synchronous, then the expected number of cluster-sending steps performed by* $\text{PCS}(\mathcal{C}_1, \mathcal{C}_2, v)$ *is upper bounded by* 4 *if* $\mathbf{n}_{\mathcal{C}_1} > 2\mathbf{f}_{\mathcal{C}_1}$ *and* $\mathbf{n}_{\mathcal{C}_2} > 2\mathbf{f}_{\mathcal{C}_2}$; *and by* $\frac{9}{4}$ $(= 2\frac{1}{4})$ *if* $\mathbf{n}_{\mathcal{C}_1} > 3\mathbf{f}_{\mathcal{C}_1}$ *and* $\mathbf{n}_{\mathcal{C}_2} > 3\mathbf{f}_{\mathcal{C}_2}$.

In PCS, the replicas $(R_1, R_2) \in \mathcal{C}_1 \times \mathcal{C}_2$ are chosen fully at random and *with replacement*, as PCS does not retain any information on *failed* probabilistic steps. In the worst case, this prevents *termination*, as the same pair of replicas can be picked repeatedly. Furthermore, PCS does not prevent the choice of faulty replicas whose failure could be detected. We can easily improve on this, as the *failure* of a probabilistic step provides some information on the chosen replicas. In specific, we have the following technical properties:

**Lemma 4.1.** *Let* $\mathcal{C}_1, \mathcal{C}_2$ *be disjoint clusters. We assume synchronous communication and assume that each replica in* $\text{nf}(\mathcal{C}_1)$ *decided* AGREE *on sending* $v$ *to* $\mathcal{C}_2$.

---

[2]Throughout this paper, the *number of consensus steps* in the presented cluster-sending protocols refers to the *single* consensus step necessary to reach agreement in the sending cluster on sending a value $v$ and all consensus steps performed in all invocations of CS-STEP by the protocol.

1. *Let* $(R_1, R_2) \in \mathcal{C}_1 \times \mathcal{C}_2$. *If* CS-STEP($R_1$, $R_2$, $v$) *fails to cluster-send* $v$, *then either* $R_1 \in f(\mathcal{C}_1)$, $R_2 \in \mathcal{C}_2$, *or both.*

2. *Let* $R_1 \in \mathcal{C}_1$. *If* CS-STEP($R_1$, $R_2$, $v$) *fails to cluster-send* $v$ *for* $\mathbf{f}_{\mathcal{C}_2} + 1$ *distinct replicas* $R_2 \in \mathcal{C}_2$, *then* $R_1 \in f(\mathcal{C}_1)$.

3. *Let* $R_2 \in \mathcal{C}_2$. *If* CS-STEP($R_1$, $R_2$, $v$) *fails to cluster-send* $v$ *for* $\mathbf{f}_{\mathcal{C}_1} + 1$ *distinct replicas* $R_1 \in \mathcal{C}_1$, *then* $R_2 \in f(\mathcal{C}_2)$.

*Proof.* The statement of this Lemma assumes that the pre-conditions for any execution of CS-STEP($R_1$, $R_2$, $v$) with $R_1 \in \mathcal{C}_1$ and $R_2 \in \mathcal{C}_2$ are established. Hence, by Proposition 3.1, CS-STEP($R_1$, $R_2$, $v$) will cluster-send $v$ if $R_1 \in nf(\mathcal{C}_1)$ and $R_2 \in nf(\mathcal{C}_2)$. If the cluster-sending step fails to cluster-send $v$, then one of the replicas involved must be faulty, proving the first property. Next, let $R_1 \in \mathcal{C}_1$ and consider a set $S \subseteq \mathcal{C}_2$ of $\mathbf{n}_S = \mathbf{f}_{\mathcal{C}_2} + 1$ replicas such that, for all $R_2 \in S$, CS-STEP($R_1$, $R_2$, $v$) fails to cluster-send $v$. Let $S' = S \setminus f(\mathcal{C}_2)$ be the non-faulty replicas in $S$. As $\mathbf{n}_S > \mathbf{f}_{\mathcal{C}_2}$, we have $\mathbf{n}_{S'} \geq 1$ and there exists a $R'_2 \in S'$. As $R'_2 \notin f(\mathcal{C}_2)$ and CS-STEP($R_1$, $R'_2$, $v$) fails to cluster-send $v$, we must have $R_1 \in f(\mathcal{C}_1)$ by the first property, proving the second property. An analogous argument proves the third property. $\square$

We can apply the properties of Lemma 4.1 to actively *prune* which replica pairs PCS considers (Line 4 of Figure 4). Notice that pruning via Lemma 4.1(1) simply replaces choosing replica pairs *with replacement*, as done by PCS, by choosing replica pairs *without replacement*, this without further reducing the possible search space. Pruning via Lemma 4.1(2) does reduce the search space, however, as each replica in $\mathcal{C}_1$ will only be paired with a subset of $\mathbf{f}_{\mathcal{C}_2} + 1$ replicas in $\mathcal{C}_2$. Likewise, pruning via Lemma 4.1(3) also reduces the search space. We obtain the *Pruned Synchronous Probabilistic Cluster-Sending protocol* (PPCS) by applying all three prune steps to PCS. By construction, Theorem 4.2, and Lemma 4.1, we conclude:

**Corollary 4.5.** *Let* $\mathcal{C}_1, \mathcal{C}_2$ *be disjoint clusters. If communication is synchronous, then* PPCS($\mathcal{C}_1$, $\mathcal{C}_2$, $v$) *results in cluster-sending* $v$ *from* $\mathcal{C}_1$ *to* $\mathcal{C}_2$. *The execution performs two local consensus steps in* $\mathcal{C}_1$, *one local consensus step in* $\mathcal{C}_2$, *is expected to make less than* $(\mathbf{n}_{\mathcal{C}_1} \mathbf{n}_{\mathcal{C}_2})/(\mathbf{nf}_{\mathcal{C}_1} \mathbf{nf}_{\mathcal{C}_2})$ *cluster-sending steps, and makes worst-case* $(\mathbf{f}_{\mathcal{C}_1} + 1)(\mathbf{f}_{\mathcal{C}_2} + 1)$ *cluster-sending steps.*

# 5 Worst-Case Linear-Time Probabilistic Cluster-Sending

In the previous section, we introduced PCS and PPCS, two probabilistic cluster-sending protocols that can cluster-send a value $v$ from $\mathcal{C}_1$ to $\mathcal{C}_2$ with expected constant cost. Unfortunately, PCS does not guarantee termination, while PPCS has a worst-case *quadratic complexity*. In this section, we improve on this by presenting a probabilistic cluster-sending protocol that has expected constant cost and guarantees termination

with a worst-case *optimal linear complexity* [18, 20]. We refer to Table 1 for an overview of the notation used in this section.

Table 1: Overview of the notation used in Section 5.

| Notation | Description |
|---|---|
| $P_1, P_2$ | Permutation of a list of replicas from $\mathcal{C}_1$ and $\mathcal{C}_2$, respectively. |
| $m_1, m_2$ | Given a pair of lists of replicas $(P_1, P_2)$, the number of faulty replicas in list $P_1$ and $P_2$, respectively. |
| $b_1$ | The number 1-*faulty pairs* in a given pair of lists of replicas $(P_1, P_2)$. |
| $b_2$ | The number 2-*faulty pairs* in a given pair of lists of replicas $(P_1, P_2)$. |
| $b_{1,2}$ | The number of *both-faulty pairs* in a given pair of lists of replicas $(P_1, P_2)$. |
| $list(R)$ | A list-representation of the replica set $R$. |
| $perms(S)$ | Permutation of list of replicas $S$. |
| $S^{:n}$ | The first $n$ elements in the list obtained by repeatedly concatenating list $S$. |
| $L\vert_M$ | Tthe list obtained from $L$ by only keeping the values that also appear in list $M$. |
| $\mathbb{M}(v, w)$ | The number of distinct ways in which two lists of $v$ and $w$ elements, respectively, can be merged together (without shuffling elements from their respective lists). |
| $\Phi$ | A list-pair function. |
| $\Vert P_1; P_2 \Vert_{\mathbf{f}}$ | The number of faulty positions in $(P_1, P_2)$. |
| $\mathbb{F}(n, m_1, m_2, k)$ | The number of permutations $(P_1, P_2)$ with $k$ faulty positions of two given lists of $n$ replicas of which $m_1$ and $m_2$ replicas are faulty, respectively. |
| $\mathbb{E}(n, m_1, m_2)$ | The non-faulty position trials problem with two lists of $n$ replicas of which $m_1$ and $m_2$ replicas are faulty, respectively. |

To improve on PCS and PPCS, we need to improve the scheme by which we select replica pairs $(R_1, R_2) \in \mathcal{C}_1 \times \mathcal{C}_2$ that we use in cluster-sending steps. The straightforward manner to guarantee a worst-case *linear complexity* is by using a scheme that can select only up-to-$n = \max(\mathbf{n}_{\mathcal{C}_1}, \mathbf{n}_{\mathcal{C}_2})$ distinct pairs $(R_1, R_2) \in \mathcal{C}_1 \times \mathcal{C}_2$. To select $n$ replica pairs from $\mathcal{C}_1 \times \mathcal{C}_2$, we will proceed in two steps.

1. We generate list $S_1$ of $n$ replicas taken from $\mathcal{C}_1$ and list $S_2$ of $n$ replicas taken from $\mathcal{C}_2$.

2. Then, we choose permutations $P_1 \in perms(S_1)$ and $P_2 \in perms(S_2)$ fully at random, and interpret each pair $(P_1[i], P_2[i])$. $0 \leq i < n$, as one of the chosen replica pairs.

We use the first step to deal with any differences in the sizes of $\mathcal{C}_1$ and $\mathcal{C}_2$, and we use the second step to introduce sufficient randomness in our protocol to yield a low expected-case message complexity.

Next, we introduce some notations to simplify reasoning about the above list-based scheme. If $R$ is a set of replicas, then $\mathsf{list}(R)$ is the list consisting of the replicas in $R$ placed in a predetermined order (e.g., on increasing replica identifier). If $S$ is a list of replicas, then we write $\mathsf{f}(S)$ to denote the faulty replicas in $S$ and $\mathsf{nf}(S)$ to denote the non-faulty replicas in $S$, and we write $\mathbf{n}_S = |S|$, $\mathbf{f}_S = |\{i \mid (0 \le i < \mathbf{n}_S) \wedge S[i] \in \mathsf{f}(S)\}|$, and $\mathbf{nf}_S = \mathbf{n}_S - \mathbf{f}_S$ to denote the number of positions in $S$ with replicas, faulty replicas, and non-faulty replicas, respectively. If $(P_1, P_2)$ is a pair of equal-length lists of $n = |P_1| = |P_2|$ replicas, then we say that the $i$-th position is a *faulty position* if either $P_1[i] \in \mathsf{f}(P_1)$ or $P_2[i] \in \mathsf{f}(P_2)$. We write $\|P_1; P_2\|_{\mathbf{f}}$ to denote the number of *faulty positions* in $(P_1, P_2)$. As faulty positions can only be constructed out of the $\mathbf{f}_{P_1}$ faulty replicas in $P_1$ and the $\mathbf{f}_{P_2}$ faulty replicas in $P_2$, we must have $\max(\mathbf{f}_{P_1}, \mathbf{f}_{P_2}) \le \|P_1; P_2\|_{\mathbf{f}} \le \min(n, \mathbf{f}_{P_1} + \mathbf{f}_{P_2})$.

*Example* 5.1. Consider clusters $\mathcal{C}_1, \mathcal{C}_2$ with

$$S_1 = \mathsf{list}(\mathcal{C}_1) = [\mathsf{R}_{1,1}, \dots, \mathsf{R}_{1,5}], \quad \mathsf{f}(\mathcal{C}_1) = \{\mathsf{R}_{1,1}, \mathsf{R}_{1,2}\};$$
$$S_2 = \mathsf{list}(\mathcal{C}_2) = [\mathsf{R}_{2,1}, \dots, \mathsf{R}_{2,5}], \quad \mathsf{f}(\mathcal{C}_2) = \{\mathsf{R}_{2,1}, \mathsf{R}_{2,2}\}.$$

The set $\mathsf{perms}(S_1) \times \mathsf{perms}(S_2)$ contains $5!^2 = 14400$ list pairs. Now, consider the list pairs $(P_1, P_2), (Q_1, Q_2), (R_1, R_2) \in \mathsf{perms}(S_1) \times \mathsf{perms}(S_2)$ with

$$P_1[\mathsf{R}_{1,1}, \mathsf{R}_{1,5}, \mathsf{R}_{1,2}, \mathsf{R}_{1,4}, \mathsf{R}_{1,3}],$$
$$P_2[\mathsf{R}_{2,1}, \mathsf{R}_{2,3}, \mathsf{R}_{2,2}, \mathsf{R}_{2,5}, \mathsf{R}_{2,4}];$$
$$Q_1[\mathsf{R}_{1,1}, \mathsf{R}_{1,3}, \mathsf{R}_{1,5}, \mathsf{R}_{1,4}, \mathsf{R}_{1,2}],$$
$$Q_2[\mathsf{R}_{2,5}, \mathsf{R}_{2,4}, \mathsf{R}_{2,3}, \mathsf{R}_{2,2}, \mathsf{R}_{2,1}];$$
$$R_1[\mathsf{R}_{1,5}, \mathsf{R}_{1,4}, \mathsf{R}_{1,3}, \mathsf{R}_{1,2}, \mathsf{R}_{1,1}],$$
$$R_2[\mathsf{R}_{2,1}, \mathsf{R}_{2,2}, \mathsf{R}_{2,3}, \mathsf{R}_{2,4}, \mathsf{R}_{2,5}].$$

We have underlined the faulty replicas in each list, and we have $\|P_1; P_2\|_{\mathbf{f}} = 2 = \mathbf{f}_{S_1} = \mathbf{f}_{S_2}$, $\|Q_1; Q_2\|_{\mathbf{f}} = 3$, and $\|R_1; R_2\|_{\mathbf{f}} = 4 = \mathbf{f}_{S_1} + \mathbf{f}_{S_2}$.

In the following, we will use a *list-pair function* $\Phi$ to compute the initial list-pair $(S_1, S_2)$ of $n$ replicas taken from $\mathcal{C}_1$ and $\mathcal{C}_2$, respectively. We build a cluster-sending protocol that uses $\Phi$ to compute $S_1$ and $S_2$, uses randomization to choose $n$ replica pairs from $S_1 \times S_2$, and, finally, performs cluster-sending steps using only these $n$ replica pairs. The pseudo-code of the resultant *Synchronous Probabilistic Linear Cluster-Sending protocol* PLCS can be found in Figure 5. Next, we prove that PLCS is correct and has a worst-case linear message complexity:

**Proposition 5.1.** *Let $\mathcal{C}_1, \mathcal{C}_2$ be disjoint clusters and let $\Phi$ be a list-pair function with $(S_1, S_2) := \Phi(\mathcal{C}_1, \mathcal{C}_2)$ and $n = \mathbf{n}_{S_1} = \mathbf{n}_{S_2}$. If communication is synchronous and $n > \mathbf{f}_{S_1} + \mathbf{f}_{S_2}$, then $\mathrm{PLCS}(\mathcal{C}_1, \mathcal{C}_2, v, \Phi)$ results in cluster-sending $v$ from $\mathcal{C}_1$ to $\mathcal{C}_2$. The execution performs two local consensus steps in $\mathcal{C}_1$, one local consensus step in $\mathcal{C}_2$, and makes worst-case $\mathbf{f}_{S_1} + \mathbf{f}_{S_2} + 1$ cluster-sending steps.*

---

**Protocol** $\mathrm{PLCS}(\mathcal{C}_1, \mathcal{C}_2, v, \Phi)$**:**

1: Use *local consensus* on $v$ and construct $\langle \mathtt{send} : v, \mathcal{C}_2 \rangle_{\mathcal{C}_1}$.
2: {*Each replica in* $\mathsf{nf}(\mathcal{C}_1)$ *decides* AGREE *on $v$.*}
3: Let $(S_1, S_2) := \Phi(\mathcal{C}_1, \mathcal{C}_2)$.
4: Choose $(P_1, P_2) \in \mathsf{perms}(S_1) \times \mathsf{perms}(S_2)$ fully at random.
5: $i := 0$.
6: **repeat**
7:     CS-STEP$(P_1[i], P_2[i], v)$
8:     Wait *three* global pulses.
9:     $i := i + 1$.
10: **until** $\mathcal{C}_1$ reaches consensus on $\langle \mathtt{proof} : \langle \mathtt{send} : v, \mathcal{C}_2 \rangle_{\mathcal{C}_1} \rangle_{\mathcal{C}_2}$.

---

Figure 5: The Synchronous Probabilistic Linear Cluster-Sending protocol $\mathrm{PLCS}(\mathcal{C}_1, \mathcal{C}_2, v, \Phi)$ that cluster-sends a value $v$ from $\mathcal{C}_1$ to $\mathcal{C}_2$ using list-pair function $\Phi$.

*Proof.* Due to Lines 1–2 of Figure 5, $\mathrm{PLCS}(\mathcal{C}_1, \mathcal{C}_2, v, \Phi)$ establishes the pre-conditions for any execution of CS-STEP$(\mathsf{R}_1, \mathsf{R}_2, v)$ with $\mathsf{R}_1 \in \mathcal{C}_1$ and $\mathsf{R}_2 \in \mathcal{C}_2$. Now let $(P_1, P_2) \in \mathsf{perms}(S_1) \times \mathsf{perms}(S_2)$, as chosen at Line 4 of Figure 5. As $P_i, i \in \{1, 2\}$, is a permutation of $S_i$, we have $\mathbf{f}_{P_i} = \mathbf{f}_{S_i}$. Hence, we have $\|P_1; P_2\|_{\mathbf{f}} \le \mathbf{f}_{S_1} + \mathbf{f}_{S_2}$ and there must exist a position $j$, $0 \le j < n$, such that $(P_1[j], P_2[j]) \in \mathsf{nf}(\mathcal{C}_1) \times \mathsf{nf}(\mathcal{C}_2)$. Using the correctness of CS-STEP (Proposition 3.1), we conclude that $\mathrm{PLCS}(\mathcal{C}_1, \mathcal{C}_2, v, \Phi)$ results in cluster-sending $v$ from $\mathcal{C}_1$ to $\mathcal{C}_2$ in at most $\mathbf{f}_{S_1} + \mathbf{f}_{S_2} + 1$ cluster-sending steps. Finally, the bounds on the number of consensus steps follow from an argument analogous to the one in the proof of Theorem 4.2. $\square$

Proposition 5.1 only shows that PLCS will perform cluster-sending when specific conditions are met on the list-pair function. Next, we proceed in two steps to arrive at practical list-pair functions for PLCS that can be used in combination with PLCS to guarantee an expected constant cost. First, in Section 5.1, we study the probabilistic nature of PLCS. Then, in Section 5.2, we propose practical list-pair functions and show that these functions yield instances of PLCS with expected constant message complexity.

## 5.1 The Expected-Case Complexity of PLCS

The expected-case analysis of PCS and PPCS was rather straightforward, as the sending and receiving replicas used by these protocols are chosen fully at random and independent of each other. Hence, the random choices made by both protocols can be modelled via well-known independent Bernoulli trials (see the proof of Theorem 4.2). In PLCS, the choice of sending and receiving replicas are dependent, as they are chosen from a list of possible replica pairs. As such, the random choices made by PLCS can no longer be modelled via independent Bernoulli trials. Hence, the expected-case analysis of PLCS requires a further analysis of the probabilistic nature of the randomized scheme used by PLCS.

As the first step toward this analysis, we solve the following abstract problem that captures the probabilistic argument at the core of the expected-case complexity of PLCS:

**Problem 5.2** (non-faulty position trials)**.** Let $S_1$ and $S_2$ be lists of $|S_1| = |S_2| = n$ replicas. Choose permutations $(P_1, P_2) \in$ perms$(S_1) \times$ perms$(S_2)$ fully at random. Next, we inspect positions in $P_1$ and $P_2$ fully at random (with replacement). The *non-faulty position trials problem* asks how many positions one expects to inspect to find the first non-faulty position.

Let $S_1$ and $S_2$ be lists of $|S_1| = |S_2| = n$ replicas. To answer the non-faulty position trials problem, we first look at the combinatorics of *faulty positions* in pairs $(P_1, P_2) \in$ perms$(S_1) \times$ perms$(S_2)$. Let $m_1 = \mathbf{f}_{S_1}$ and $m_2 = \mathbf{f}_{S_2}$. By $\mathbb{F}(n, m_1, m_2, k)$, we denote the number of distinct pairs $(P_1, P_2)$ one can construct that have exactly $k$ faulty positions, hence, with $\|P_1; P_2\|_{\mathbf{f}} = k$. As observed, we have $\max(m_1, m_2) \leq \|P_1; P_2\|_{\mathbf{f}} \leq \min(n, m_1 + m_2)$ for any pair $(P_1, P_2)$. Hence, we have $\mathbb{F}(n, m_1, m_2, k) = 0$ for all $k < \max(m_1, m_2)$ and $k > \min(n, m_1 + m_2)$.

Now consider the step-wise construction of any permutation $(P_1, P_2) \in$ perms$(S_1) \times$ perms$(S_2)$ with $k$ faulty positions. First, we choose $(P_1[0], P_2[0])$, the pair at position 0, after which we choose pairs for the remaining $n - 1$ positions. For $P_i[0]$, $i \in \{1, 2\}$, we can choose $n$ distinct replicas, of which $m_i$ are faulty. If we pick a non-faulty replica, then the remainder of $P_i$ is constructed out of $n - 1$ replicas, of which $m_i$ are faulty. Otherwise, the remainder of $P_i$ is constructed out of $n - 1$ replicas of which $m_i - 1$ are faulty. If, due to our choice of $(P_1[0], P_2[0])$, the first position is faulty, then only $k - 1$ out of the $n - 1$ remaining positions must be faulty. Otherwise, $k$ out of the $n - 1$ remaining positions must be faulty. Combining this analysis yields four types for the first pair $(P_1[0], P_2[0])$:

1. A *non-faulty pair* $(P_1[0], P_2[0]) \in$ nf$(P_1) \times$ nf$(P_2)$. We have $(n - m_1)(n - m_2)$ such pairs, and we have $\mathbb{F}(n - 1, m_1, m_2, k)$ different ways to construct the remainder of $P_1$ and $P_2$.

2. A 1-*faulty pair* $(P_1[0], P_2[0]) \in$ f$(P_1) \times$ nf$(P_2)$. We have $m_1(n - m_2)$ such pairs, and we have $\mathbb{F}(n - 1, m_1 - 1, m_2, k - 1)$ different ways to construct the remainder of $P_1$ and $P_2$.

3. A 2-*faulty pair* $(P_1[0], P_2[0]) \in$ nf$(P_1) \times$ f$(P_2)$. We have $(n - m_1)m_2$ such pairs, and we have $\mathbb{F}(n - 1, m_1, m_2 - 2, k - 1)$ different ways to construct the remainder of $P_1$ and $P_2$.

4. A *both-faulty pair* $(P_1[0], P_2[0]) \in$ f$(P_1) \times$ f$(P_2)$. We have $m_1 m_2$ such pairs, and we have $\mathbb{F}(n - 1, m_1 - 1, m_2 - 1, k - 1)$ different ways to construct the remainder of $P_1$ and $P_2$.

Hence, for all $k$, $\max(m_1, m_2) \leq k \leq \min(n, m_1 + m_2)$, $\mathbb{F}(n, m_1, m_2, k)$ is recursively defined by:

$$\mathbb{F}(n, m_1, m_2, k) = (n - m_1)(n - m_2)\mathbb{F}(n - 1, m_1, m_2, k)$$
$$\text{(non-faulty pair)}$$
$$+ m_1(n - m_2)\mathbb{F}(n - 1, m_1 - 1, m_2, k - 1)$$
$$\text{(1-faulty pair)}$$
$$+ (n - m_1)m_2\mathbb{F}(n - 1, m_1, m_2 - 1, k - 1)$$
$$\text{(2-faulty pair)}$$
$$+ m_1 m_2 \mathbb{F}(n - 1, m_1 - 1, m_2 - 1, k - 1),$$
$$\text{(both-faulty pair)}$$

and the base case for this recursion is $\mathbb{F}(0, 0, 0, 0) = 1$.

*Example* 5.3. Reconsider the list pairs $(P_1, P_2)$, $(Q_1, Q_2)$, and $(R_1, R_2)$ from Example 5.1. In $(P_1, P_2)$, we have both-faulty pairs at positions 0 and 2 and non-faulty pairs at positions 1, 3, and 4. In $(Q_1, Q_2)$, we have a 1-faulty pair at position 0, non-faulty pairs at positions 1 and 2, a 2-faulty pair at position 3, and a both-faulty pair at position 4. Finally, in $(R_1, R_2)$, we have 2-faulty pairs at positions 0 and 1, a non-faulty pair at position 2, and 1-faulty pairs at positions 3 and 4.

Using the combinatorics of faulty positions, we formalize an exact solution to the *non-faulty position trials problem*:

**Lemma 5.1.** *Let $S_1$ and $S_2$ be lists of $n = \mathbf{n}_{S_1} = \mathbf{n}_{S_2}$ replicas with $m_1 = \mathbf{f}_{S_1}$ and $m_2 = \mathbf{f}_{S_2}$. If $m_1 + m_2 < n$, then the non-faulty position trials problem $\mathbb{E}(n, m_1, m_2)$ has solution*

$$\frac{1}{n!^2}\left(\sum_{k=\max(m_1, m_2)}^{m_1 + m_2} \frac{n}{n - k}\mathbb{F}(n, m_1, m_2, k)\right).$$

*Proof.* We have $|\text{perms}(S_1)| = |\text{perms}(S_2)| = n!$. Consequently, we have $|\text{perms}(S_1) \times \text{perms}(S_2)| = n!^2$ and we have probability $1/(n!^2)$ to choose any pair $(P_1, P_2) \in$ perms$(S_1) \times$ perms$(S_2)$. Now consider such a pair $(P_1, P_2) \in$ perms$(S_1) \times$ perms$(S_2)$. As there are $\|P_1; P_2\|_{\mathbf{f}}$ faulty positions in $(P_1, P_2)$, we have probability $p(P_1, P_2) = (n - \|P_1; P_2\|_{\mathbf{f}})/n$ to inspect a non-faulty position. Notice that $\max(m_1, m_2) \leq \|P_1; P_2\|_{\mathbf{f}} \leq m_1 + m_2 < n$ and, hence, $0 < p(P_1, P_2) \leq 1$. Each of the inspected positions in $(P_1, P_2)$ is chosen fully at random. Hence, each inspection is a *Bernoulli trial* with probability of success $p(P_1, P_2)$, and we expect to inspect a first non-faulty position in the $p(P_1, P_2)^{-1} = n/(n - \|P_1; P_2\|_{\mathbf{f}})$-th attempt. We conclude that the non-faulty position trials problem $\mathbb{E}(n, m_1, m_2)$ has solution

$$\frac{1}{n!^2}\left(\sum_{(P_1, P_2) \in \text{perms}(S_1) \times \text{perms}(S_2)} \frac{n}{n - \|P_1; P_2\|_{\mathbf{f}}}\right).$$

Notice that there are $\mathbb{F}(n, m_1, m_2, k)$ distinct pairs $(P_1, P_2) \in$ perms$(S_1) \times$ perms$(S_2)$ with $\|P_1'; P_2'\|_{\mathbf{f}} = k$ for each $k$, $\max(m_1, m_2) \leq k \leq m_1 + m_2 < n$. Hence, in the above expression for $\mathbb{E}(n, m_1, m_2)$, we can group on these pairs $(P_1', P_2')$ to obtain the searched-for solution. $\square$

To further solve the non-faulty position trials problem, we work towards a *closed form* for $\mathbb{F}(n, m_1, m_2, k)$. Consider any pair $(P_1, P_2) \in \text{perms}(S_1) \times \text{perms}(S_2)$ with $\|P_1; P_2\|_{\mathbf{f}} = k$ obtained via the outlined step-wise construction. Let $b_1$ be the number of 1-*faulty pairs*, let $b_2$ be the number of 2-*faulty pairs*, and let $b_{1,2}$ be the number of *both-faulty pairs* in $(P_1, P_2)$. By construction, we must have $k = b_1 + b_2 + b_{1,2}$, $m_1 = b_1 + b_{1,2}$, and $m_2 = b_2 + b_{1,2}$ and by rearranging terms, we can derive

$$b_{1,2} = (m_1 + m_2) - k, \quad b_1 = k - m_2, \quad b_2 = k - m_1.$$

*Example* 5.4. Consider

$$\begin{aligned} S_1 &= [R_{1,1}, \ldots, R_{1,5}], & f(S_1) &= \{R_{1,1}, R_{1,2}, R_{1,3}\}; \\ S_2 &= [R_{2,1}, \ldots, R_{2,5}], & f(S_2) &= \{R_{2,1}\}. \end{aligned}$$

Hence, we have $n = 5$, $m_1 = \mathbf{f}_{S_1} = 3$, and $m_2 = \mathbf{f}_{S_2} = 1$. If we want to create a pair $(P_1, P_2) \in \text{perms}(S_1) \times \text{perms}(S_2)$ with $k = \|P_1; P_2\|_{\mathbf{f}} = 3$ faulty positions, then $(P_1, P_2)$ must have two non-faulty pairs, two 1-faulty pairs, no 2-faulty pairs, and one both-faulty pair. Hence, we have $n - k = 2$, $b_1 = 2$, $b_2 = 0$, and $b_{1,2} = 1$.

The above analysis only depends on the choice of $m_1$, $m_2$, and $k$, and not on our choice of $(P_1, P_2)$. Next, we use this analysis to express $\mathbb{F}(n, m_1, m_2, k)$ in terms of the number of distinct ways in which one can *construct*

(A) lists of $b_1$ 1-faulty pairs out of faulty replicas from $S_1$ and non-faulty replicas from $S_2$,

(B) lists of $b_2$ 2-faulty pairs out of non-faulty replicas from $S_1$ and faulty replicas from $S_2$,

(C) lists of $b_{1,2}$ both-faulty pairs out of the remaining faulty replicas in $S_1$ and $S_2$ that are not used in the previous two cases, and

(D) lists of $n - k$ non-faulty pairs out of the remaining (non-faulty) replicas in $S_1$ and $S_2$ that are not used in the previous three cases;

and in terms of the number of distinct ways one can *merge* these lists. As the first step, we look at how many distinct ways we can merge two lists together:

**Lemma 5.2.** *For any two disjoint lists $S$ and $T$ with $|S| = v$ and $|T| = w$, there exist $\mathbb{M}(v, w) = (v + w)!/(v!w!)$ distinct lists $L$ with $L|_S = S$ and $L|_T = T$, in which $L|_M$, $M \in \{S, T\}$, is the list obtained from $L$ by only keeping the values that also appear in list $M$.*

Next, we look at the number of distinct ways in which one can construct lists of type A, B, C, and D. Consider the construction of a list of type A. We can choose $\binom{m_1}{b_1}$ distinct sets of $b_1$ faulty replicas from $S_1$ and we can choose $\binom{n-m_2}{b_1}$ distinct sets of $b_1$ non-faulty replicas from $S_2$. As we

can order the chosen values from $S_1$ and $S_2$ in $b_1!$ distinct ways, we can construct $b_1!^2 \binom{m_1}{b_1} \binom{n-m_2}{b_1}$ distinct lists of type A. Likewise, we can construct $b_2!^2 \binom{n-m_1}{b_2} \binom{m_2}{b_2}$ distinct lists of type B.

*Example* 5.5. We continue from the setting of Example 5.4: we want to create a pair $(P_1, P_2) \in \text{perms}(S_1) \times \text{perms}(S_2)$ with $k = \|P_1; P_2\|_{\mathbf{f}} = 3$ faulty positions. To create $(P_1, P_2)$, we need to create $b_1 = 2$ pairs that are 1-faulty. We have $\binom{m_1}{b_1} = \binom{3}{2} = 3$ sets of two faulty replicas in $S_1$ that we can choose, namely the sets $\{R_{1,1}, R_{1,2}\}$, $\{R_{1,1}, R_{1,3}\}$, and $\{R_{1,2}, R_{1,3}\}$. Likewise, we have $\binom{n-m_2}{b_1} = \binom{4}{2} = 6$ sets of two non-faulty replicas in $S_2$ that we can choose. Assume we choose $T_1 = \{R_{1,1}, R_{1,3}\}$ from $S_1$ and $T_2 = \{R_{2,4}, R_{2,5}\}$ from $S_2$. The two replicas in $T_1$ can be ordered in $\mathbf{n}_{T_1}! = 2! = 2$ ways, namely $[R_{1,1}, R_{1,3}]$ and $[R_{1,3}, R_{1,1}]$. Likewise, the two replicas in $T_2$ can be ordered in $\mathbf{n}_{T_2}! = 2! = 2$ ways. Hence, we can construct $2 \cdot 2 = 4$ distinct lists of type A out of this single choice for $T_1$ and $T_2$, and the sequences $S_1$ and $S_2$ provide us with $\binom{m_1}{b_1} \binom{n-m_2}{b_1} = 18$ distinct choices for $T_1$ and $T_2$. We conclude that we can construct 72 distinct lists of type A from $S_1$ and $S_2$.

By construction, lists of type A and type B cannot utilize the same replicas from $S_1$ or $S_2$. After choosing $b_1 + b_2$ replicas in $S_1$ and $S_2$ for the construction of lists of type A and B, the remaining $b_{1,2}$ faulty replicas in $S_1$ and $S_2$ are all used for constructing lists of type C. As we can order these remaining values from $S_1$ and $S_2$ in $b_{1,2}!$ distinct ways, we can construct $b_{1,2}!^2$ distinct lists of type C (per choice of lists of type A and B). Likewise, the remaining $n - k$ non-faulty replicas in $S_1$ and $S_2$ are all used for constructing lists of type D, and we can construct $(n - k)!^2$ distinct lists of type D (per choice of lists of type A and B).

As the final steps, we merge lists of type A and B into lists of type AB. We can do so in $\mathbb{M}(b_1, b_2)$ ways and the resultant lists have size $b_1 + b_2$. Next, we merge lists of type AB and C into lists of type ABC. We can do so in $\mathbb{M}(b_1 + b_2, b_{1,2})$ ways and the resultant lists have size $k$. Finally, we merge list of type ABC and D together, which we can do in $\mathbb{M}(k, n - k)$ ways. From this construction, we derive that $\mathbb{F}(n, m_1, m_2, k)$ is equivalent to

$$b_1!^2 \binom{m_1}{b_1} \binom{n-m_2}{b_1} b_2!^2 \binom{n-m_1}{b_2} \binom{m_2}{b_2} \cdot$$
$$\mathbb{M}(b_1, b_2) b_{1,2}!^2 \mathbb{M}(b_1 + b_2, b_{1,2})(n-k)!^2 \mathbb{M}(k, n-k),$$

which can be simplified to the following (see Appendix B):

**Lemma 5.3.** *Let $\max(m_1, m_2) \le k \le \min(n, m_1 + m_2)$ and let $b_1 = k - m_2$, $b_2 = k - m_1$, and $b_{1,2} = (m_1 + m_2) - k$. We have*

$$\mathbb{F}(n, m_1, m_2, k) = \frac{m_1! m_2! (n - m_1)! (n - m_2)! n!}{b_1! b_2! b_{1,2}! (n - k)!}.$$

*Proof.* We write $f(n,m_1,m_2,k)$ for the closed form in the statement of this lemma and we prove the statement of this lemma by induction. First, the base case $\mathbb{F}(0,0,0,0)$. In this case, we have $n = m_1 = m_2 = k = 0$ and, hence, $b_1 = b_2 = b_{1,2} = 0$, and we conclude $f(0,0,0,0) = 1 = \mathbb{F}(0,0,0,0)$.

Now assume $\mathbb{F}(n',m_1',m_2',k') = f(n',m_1',m_2',k')$ for all $n' < n$ and all $k'$ with $\max(m_1',m_2') \le k' \le \min(n',m_1' + m_2')$. Next, we prove $\mathbb{F}(n,m_1,m_2,k) = f(n,m_1,m_2,k)$ with $\max(m_1,m_2) \le k \le \min(n,m_1 + m_2)$. We use the shorthand $\mathbb{G} = \mathbb{F}(n,m_1,m_2,k)$ and we have

$$\mathbb{G} = (n-m_1)(n-m_2)\mathbb{F}(n-1,m_1,m_2,k)$$
$$\text{(non-faulty pair)}$$
$$+ m_1(n-m_2)\mathbb{F}(n-1,m_1-1,m_2,k-1)$$
$$\text{(1-faulty pair)}$$
$$+ (n-m_1)m_2\mathbb{F}(n-1,m_1,m_2-1,k-1)$$
$$\text{(2-faulty pair)}$$
$$+ m_1 m_2 \mathbb{F}(n-1,m_1-1,m_2-1,k-1).$$
$$\text{(both-faulty pair)}$$

Notice that if $n = k$, then the non-faulty pair case does not apply, as $\mathbb{F}(n-1,m_1,m_2,k) = 0$, and evaluates to zero. Likewise, if $b_1 = 0$, then the 1-faulty pair case does not apply, as $\mathbb{F}(n-1,m_1-1,m_2,k-1) = 0$, and evaluates to zero; if $b_2 = 0$, then the 2-faulty pair case does not apply, as $\mathbb{F}(n-1,m_1,m_2-1,k-1) = 0$, and evaluates to zero; and, finally, if $b_{1,2} = 0$, then the both-faulty pair case does not apply, as $\mathbb{F}(n-1,m_1-1,m_2-1,k-1) = 0$, and evaluates to zero.

First, we consider the case in which $n > k$, $b_1 > 0$, $b_2 > 0$, and $b_{1,2} > 0$. Hence, each of the four cases apply and evaluate to non-zero values. We directly apply the induction hypothesis on $\mathbb{F}(n-1,m_1,m_2,k)$, $\mathbb{F}(n-1,m_1-1,m_2,k-1)$, $\mathbb{F}(n-1,m_1,m_2-1,k-1)$, and $\mathbb{F}(n-1,m_1-1,m_2-1,k-1)$, and obtain

$$\mathbb{G} = (n-m_1)(n-m_2) \cdot$$
$$\frac{m_1!m_2!(n-1-m_1)!(n-1-m_2)!(n-1)!}{b_1!b_2!b_{1,2}!(n-1-k)!}$$
$$+ m_1(n-m_2) \cdot$$
$$\frac{(m_1-1)!m_2!(n-m_1)!(n-1-m_2)!(n-1)!}{(b_1-1)!b_2!b_{1,2}!(n-1-(k-1))!}$$
$$+ (n-m_1)m_2 \cdot$$
$$\frac{m_1!(m_2-1)!(n-1-m_1)!(n-m_2)!(n-1)!}{b_1!(b_2-1)!b_{1,2}!(n-1-(k-1))!}$$
$$+ m_1 m_2 \cdot$$
$$\frac{(m_1-1)!(m_2-1)!(n-m_1)!(n-m_2)!(n-1)!}{b_1!b_2!(b_{1,2}-1)!(n-1-(k-1))!}.$$

We apply $x! = x(x-1)!$ and further simplify and obtain

$$\mathbb{G} = \frac{m_1!m_2!(n-m_1)!(n-m_2)!(n-1)!}{b_1!b_2!b_{1,2}!(n-1-k)!}$$
$$+ \frac{m_1!m_2!(n-m_1)!(n-m_2)!(n-1)!}{(b_1-1)!b_2!b_{1,2}!(n-k)!}$$
$$+ \frac{m_1!m_2!(n-m_1)!(n-m_2)!(n-1)!}{b_1!(b_2-1)!b_{1,2}!(n-k)!}$$
$$+ \frac{m_1!m_2!(n-m_1)!(n-m_2)!(n-1)!}{b_1!b_2!(b_{1,2}-1)!(n-k)!}$$
$$= (n-k)\frac{m_1!m_2!(n-m_1)!(n-m_2)!(n-1)!}{b_1!b_2!b_{1,2}!(n-k)!}$$
$$+ b_1 \frac{m_1!m_2!(n-m_1)!(n-m_2)!(n-1)!}{b_1!b_2!b_{1,2}!(n-k)!}$$
$$+ b_2 \frac{m_1!m_2!(n-m_1)!(n-m_2)!(n-1)!}{b_1!b_2!b_{1,2}!(n-k)!}$$
$$+ b_{1,2} \frac{m-1!m_2!(n-m_1)!(n-m_2)!(n-1)!}{b_1!b_2!b_{1,2}!(n-k)!}.$$

We have $k = b_1 + b_2 + b_{1,2}$ and, hence, $n = (n-k) + b_1 + b_2 + b_{1,2}$ and we conclude

$$\mathbb{G} = ((n-k) + b_1 + b_2 + b_{1,2}) \cdot$$
$$\frac{m_1!m_2!(n-m_1)!(n-m_2)!(n-1)!}{b_1!b_2!b_{1,2}!(n-k)!}$$
$$= n\frac{m_1!m_2!(n-m_1)!(n-m_2)!(n-1)!}{b_1!b_2!b_{1,2}!(n-k)!}$$
$$= \frac{m_1!m_2!(n-m_1)!(n-m_2)!n!}{b_1!b_2!b_{1,2}!(n-k)!}.$$

Next, in all other cases, we can repeat the above derivation while removing the terms corresponding to the cases that evaluate to 0. By doing so, we end up with the expression

$$\mathbb{G} = \frac{((\sum_{t \in T} t)\, m_1!m_2!(n-m_1)!(n-m_2)!(n-1)!}{b_1!b_2!b_{1,2}!(n-k)!}.$$

in which $T$ contains the term $(n-k)$ if $n > k$ (the non-faulty pair case applies), the term $b_1$ if $b_1 > 0$ (the 1-faulty case applies), the term $b_2$ if $b_2 > 0$ (the 2-faulty case applies), and the term $b_{1,2}$ if $b_{1,2} > 0$ (the both-faulty case applies). As each term $(n-k)$, $b_1$, $b_2$, and $b_{1,2}$ is in $T$ whenever the term is non-zero, we have $\sum_{t \in T} t = (n-k) + b_1 + b_2 + b_{1,2} = n$. Hence, we can repeat the steps of the above derivation in all cases, and complete the proof. □

We combine Lemma 5.1 and Lemma 5.3 to conclude

**Proposition 5.2.** *Let $S_1$ and $S_2$ be lists of $n = \mathbf{n}_{S_1} = \mathbf{n}_{S_2}$ replicas with $m_1 = \mathbf{f}_{S_1}$, $m_2 = \mathbf{f}_{S_2}$, $b_1 = k - m_2$, $b_2 = k - m_1$, and $b_{1,2} = (m_1 + m_2) - k$. If $m_1 + m_2 < n$, then the non-faulty*

*position trials problem* $\mathbb{E}(n, m_1, m_2)$ *has solution*

$$\frac{1}{n!^2}\left(\sum_{k=\max(m_1,m_2)}^{m_1+m_2} \frac{n}{n-k}\frac{m_1!m_2!(n-m_1)!(n-m_2)!n!}{b_1!b_2!b_{1,2}!(n-k)!}\right).$$

Finally, we use Proposition 5.2 to derive

**Proposition 5.3.** *Let $C_1, C_2$ be disjoint clusters and let $\Phi$ be a list-pair function with $(S_1, S_2) := \Phi(C_1, C_2)$ and $n = \mathbf{n}_{S_1} = \mathbf{n}_{S_2}$. If communication is synchronous and $\mathbf{f}_{S_1} + \mathbf{f}_{S_2} < n$, then the expected number of cluster-sending steps performed by* $\textsc{Plcs}(C_1, C_2, v, \Phi)$ *is less than* $\mathbb{E}(n, \mathbf{f}_{S_1}, \mathbf{f}_{S_2})$.

*Proof.* Let $(P_1, P_2) \in \mathsf{perms}(S_1) \times \mathsf{perms}(S_2)$. We notice that $\textsc{Plcs}$ inspects positions in $P_1$ and $P_2$ in a different way than the non-faulty trials problem: at Line 7 of Figure 5, positions are inspected one-by-one in a predetermined order and not fully at random (with replacement). Next, we will argue that $\mathbb{E}(n, \mathbf{f}_{S_1}, \mathbf{f}_{S_2})$ provides an upper bound on the expected number of cluster-sending steps regardless of these differences. Without loss of generality, we assume that $S_1$ and $S_2$ each have $n$ distinct replicas. Consequently, the pair $(P_1, P_2)$ represents a set $R$ of $n$ distinct replica pairs taken from $C_1 \times C_2$. We notice that each of the $n!$ permutations of $R$ is represented by a single pair $(P_1', P_2') \in \mathsf{perms}(S_1) \times \mathsf{perms}(S_2)$.

Now consider the selection of positions in $(P_1, P_2)$ fully at random, but without replacement. This process will yield a list $[j_0, \ldots, j_{n-1}] \in \mathsf{perms}([0, \ldots, n-1])$ of positions fully at random. Let $Q_i = [P_i[j_0], \ldots, P_i[j_{n-1}]]$, $i \in \{1, 2\}$. We notice that the pair $(Q_1, Q_2)$ also represents $R$ and we have $(Q_1, Q_2) \in \mathsf{perms}(S_1) \times \mathsf{perms}(S_2)$. Hence, by choosing a pair $(P_1, P_2) \in \mathsf{perms}(S_1) \times \mathsf{perms}(S_2)$, we choose set $R$ fully at random and, at the same time, we choose the order in which replica pairs in $R$ are inspected fully at random.

Finally, we note that $\textsc{Plcs}$ inspects positions without replacement. As the number of expected positions inspected in the non-faulty position trials problem decreases if we choose positions without replacement, we have proven that $\mathbb{E}(n, \mathbf{f}_{S_1}, \mathbf{f}_{S_2})$ is an upper bound on the expected number of cluster-sending steps. □

## 5.2 Practical Instances of $\textsc{Plcs}$

As the last step in providing practical instances of $\textsc{Plcs}$, we need to provide practical list-pair functions to be used in conjunction with $\textsc{Plcs}$. We provide two such functions that address most practical environments. Let $C_1, C_2$ be disjoint clusters, let $n_{\min} = \min(\mathbf{n}_{C_1}, \mathbf{n}_{C_2})$, and let $n_{\max} = \max(\mathbf{n}_{C_1}, \mathbf{n}_{C_2})$. We provide list-pair functions

$$\Phi_{\min}(C_1, C_2) \mapsto (\mathsf{list}(C_1)^{:n_{\min}}, \mathsf{list}(C_2)^{:n_{\min}}),$$
$$\Phi_{\max}(C_1, C_2) \mapsto (\mathsf{list}(C_2)^{:n_{\max}}, \mathsf{list}(C_2)^{:n_{\max}}),$$

in which $L^{:n}$ denotes the first $n$ values in the list obtained by repeating list $L$. Next, we illustrate usage of these functions:

*Example* 5.6. Consider clusters $C_1, C_2$ with

$$S_1 = \mathsf{list}(C_1) = [\mathsf{R}_{1,1}, \ldots, \mathsf{R}_{1,9}];$$
$$S_2 = \mathsf{list}(C_2) = [\mathsf{R}_{2,1}, \ldots, \mathsf{R}_{2,4}].$$

We have

$$\Phi_{\min}(C_1, C_2) = ([\mathsf{R}_{1,1}, \ldots, \mathsf{R}_{1,4}], \mathsf{list}(C_2));$$
$$\Phi_{\max}(C_1, C_2) = (\mathsf{list}(C_1), [\mathsf{R}_{2,1}, \ldots, \mathsf{R}_{2,4}, \mathsf{R}_{2,1}, \ldots, \mathsf{R}_{2,4}, \mathsf{R}_{2,1}]).$$

Next, we combine $\Phi_{\min}$ and $\Phi_{\max}$ with $\textsc{Plcs}$, show that in practical environments $\Phi_{\min}$ and $\Phi_{\max}$ satisfy the requirements put on list-pair functions in Proposition 5.1 to guarantee termination and cluster-sending, and use these results to determine the expected constant complexity of the resulting instances of $\textsc{Plcs}$.

**Theorem 5.7.** *Let $C_1, C_2$ be disjoint clusters with synchronous communication.*

1. *If $n = \min(\mathbf{n}_{C_1}, \mathbf{n}_{C_2}) > 2\max(\mathbf{f}_{C_1}, \mathbf{f}_{C_2})$, then the expected number of cluster-sending steps performed by $\textsc{Plcs}(C_1, C_2, v, \Phi_{\min})$ is upper bounded by 4. For every $(S_1, S_2) := \Phi_{\min}(C_1, C_2)$, we have $n = \mathbf{n}_{S_1} = \mathbf{n}_{S_2}$, $n > 2\mathbf{f}_{S_1}$, $n > 2\mathbf{f}_{S_2}$, and $n > \mathbf{f}_{S_1} + \mathbf{f}_{S_2}$*

2. *If $n = \min(\mathbf{n}_{C_1}, \mathbf{n}_{C_2}) > 3\max(\mathbf{f}_{C_1}, \mathbf{f}_{C_2})$, then the expected number of cluster-sending steps performed by $\textsc{Plcs}(C_1, C_2, v, \Phi_{\min})$ is upper bounded by $\frac{9}{4}$ $(= 2\frac{1}{4})$. For every $(S_1, S_2) := \Phi_{\min}(C_1, C_2)$, we have $n = \mathbf{n}_{S_1} = \mathbf{n}_{S_2}$, $n > 3\mathbf{f}_{S_1}$, $n > 3\mathbf{f}_{S_2}$, and $n > \mathbf{f}_{S_1} + \mathbf{f}_{S_2}$.*

3. *If $\mathbf{n}_{C_1} > 3\mathbf{f}_{C_1}$ and $\mathbf{n}_{C_2} > 3\mathbf{f}_{C_2}$, then the expected number of cluster-sending steps performed by $\textsc{Plcs}(C_1, C_2, v, \Phi_{\max})$ is upper bounded by 3. For every $(S_1, S_2) := \Phi_{\max}(C_1, C_2)$, we have $n = \mathbf{n}_{S_1} = \mathbf{n}_{S_2} = \max(\mathbf{n}_{C_1}, \mathbf{n}_{C_2}) > \mathbf{f}_{S_1} + \mathbf{f}_{S_2}$ and either we have $\mathbf{n}_{C_1} \geq \mathbf{n}_{C_2}$, $n > 3\mathbf{f}_{S_1}$, and $n > 2\mathbf{f}_{S_2}$; or we have $\mathbf{n}_{C_2} \geq \mathbf{n}_{C_1}$, $n > 2\mathbf{f}_{S_1}$, and $n > 3\mathbf{f}_{S_2}$.*

*Each of these instance of* $\textsc{Plcs}$ *results in cluster-sending $v$ from $C_1$ to $C_2$.*

*Proof.* First, we prove the properties of $\Phi_{\min}$ and $\Phi_{\max}$ claimed in the three statements of the theorem. In the first and second statement of the theorem, we have $\min(\mathbf{n}_{C_1}, \mathbf{n}_{C_2}) > c\max(\mathbf{f}_{C_1}, \mathbf{f}_{C_2})$, $c \in \{2, 3\}$. Let $(S_1, S_2) := \Phi_{\min}(C_1, C_2)$ and $n = \mathbf{n}_{S_1} = \mathbf{n}_{S_2}$. By definition of $\Phi_{\min}$, we have $n = \min(\mathbf{n}_{C_1}, \mathbf{n}_{C_2})$, in which case $S_i$, $i \in \{1, 2\}$, holds $n$ distinct replicas from $C_i$. Hence, we have $\mathbf{f}_{C_i} \geq \mathbf{f}_{S_i}$ and, as $n > c\max(\mathbf{f}_{C_1}, \mathbf{f}_{C_2}) \geq c\mathbf{f}_{C_i}$, also $n > c\mathbf{f}_{S_i}$. Finally, as $n > 2\mathbf{f}_{S_1}$ and $n > 2\mathbf{f}_{S_2}$, also $2n > 2\mathbf{f}_{S_1} + 2\mathbf{f}_{S_2}$ and $n > \mathbf{f}_{S_1} + \mathbf{f}_{S_2}$ holds.

In the last statement of the theorem, we have $\mathbf{n}_{C_1} > 3\mathbf{f}_{C_1}$ and $\mathbf{n}_{C_2} > 3\mathbf{f}_{C_2}$. Without loss of generality, we assume $\mathbf{n}_{C_1} \geq \mathbf{n}_{C_2}$. Let $(S_1, S_2) := \Phi_{\max}(C_1, C_2)$ and $n = \mathbf{n}_{S_1} = \mathbf{n}_{S_2}$. By definition of $\Phi_{\max}$, we have $n = \max(\mathbf{n}_{C_1}, \mathbf{n}_{C_2}) = \mathbf{n}_{C_1}$. As $n = \mathbf{n}_{C_1}$, we have $S_1 = \mathsf{list}(C_1)$. Consequently, we also have

$\mathbf{f}_{S_1} = \mathbf{f}_{C_1}$ and, hence, $\mathbf{n}_{S_1} > 3\mathbf{f}_{C_1}$. Next, we will show that $\mathbf{n}_{S_2} > 2\mathbf{f}_{S_2}$. Let $q = \mathbf{n}_{C_1} \operatorname{div} \mathbf{n}_{C_2}$ and $r = \mathbf{n}_{C_1} \bmod \mathbf{n}_{C_2}$. We note that $\operatorname{list}(C_2)^{:n}$ contains $q$ full copies of $\operatorname{list}(C_2)$ and one partial copy of $\operatorname{list}(C_2)$. Let $T \subset C_2$ be the set of replicas in this partial copy. By construction, we have $\mathbf{n}_{S_2} = q\mathbf{n}_{C_2} + r > q3\mathbf{f}_{C_2} + \mathbf{f}_T + \mathbf{nf}_T$ and $\mathbf{f}_{S_2} = q\mathbf{f}_{C_2} + \mathbf{f}_T$ with $\mathbf{f}_T \leq \min(\mathbf{f}_{C_2}, r)$. As $q > 1$ and $\mathbf{f}_{C_2} \geq \mathbf{f}_T$, we have $q\mathbf{f}_{C_2} \geq \mathbf{f}_{C_2} \geq \mathbf{f}_T$. Hence, $\mathbf{n}_{S_2} > 3q\mathbf{f}_{C_2} + \mathbf{f}_T + \mathbf{nf}_T > 2q\mathbf{f}_{C_2} + \mathbf{f}_{C_2} + \mathbf{f}_T + \mathbf{nf}_T \geq 2(q\mathbf{f}_{C_2} + \mathbf{f}_T) + \mathbf{nf}_T \geq 2\mathbf{f}_{S_2}$. Finally, as $n > 3\mathbf{f}_{S_1}$ and $n > 2\mathbf{f}_{S_2}$, also $2n > 3\mathbf{f}_{S_1} + 2\mathbf{f}_{S_2}$ and $n > \mathbf{f}_{S_1} + \mathbf{f}_{S_2}$ holds.

Now, we prove the upper bounds on the expected number of cluster-sending steps for $\operatorname{PLCS}(C_1, C_2, v, \Phi_{\min})$ with $\min(\mathbf{n}_{C_1}, \mathbf{n}_{C_2}) > 2\max(\mathbf{f}_{C_1}, \mathbf{f}_{C_2})$. By Proposition 5.3, the expected number of cluster-sending steps is upper bounded by $\mathbb{E}(n, \mathbf{f}_{S_1}, \mathbf{f}_{S_2})$. In the worst case, we have $n = 2f + 1$ with $f = \mathbf{f}_{S_1} = \mathbf{f}_{S_2}$. Hence, the expected number of cluster-sending steps is upper bounded by $\mathbb{E}(2f + 1, f, f)$, $f \geq 0$. We claim that $\mathbb{E}(2f + 1, f, f)$ simplifies to $\mathbb{E}(2f + 1, f, f) = 4 - 2/(f + 1) - f!^2/(2f)!$. Hence, for all $S_1$ and $S_2$, we have $\mathbb{E}(n, \mathbf{f}_{S_1}, \mathbf{f}_{S_2}) < 4$. An analogous argument can be used to prove the other upper bounds. □

Note that the third case of Theorem 5.7 corresponds to cluster-sending between arbitrary-sized resilient clusters that each operate using Byzantine fault-tolerant consensus protocols.

*Remark* 5.8. The upper bounds on the expected-case complexity of instances of PLCS presented in Theorem 5.7 match the upper bounds for PCS presented in Corollary 4.4. This does not imply that the expected-case complexity for these protocols is the same, however, as the probability distributions that yield these expected-case complexities are very different. To see this, consider a system in which all clusters have $n$ replicas of which $f$, $n = 2f + 1$, are faulty. Next, we denote the expected number of cluster-sending steps of protocol $P$ by $\mathbf{E}_P$, and we have

$$\mathbf{E}_{\operatorname{PCS}} = \frac{(2f + 1)^2}{(f + 1)^2} = 4 - \frac{4f + 3}{(f + 1)^2};$$

$$\mathbf{E}_{\operatorname{PLCS}} = \mathbb{E}(2f + 1, f, f) = 4 - \frac{2}{(f + 1)} - \frac{f!^2}{(2f)!}.$$

In Figure 6, we have illustrated this difference by plotting the expected-case complexity of PCS and PLCS for systems with equal-sized clusters. In practice, we see that the expected-case complexity for PCS is slightly lower than the expected-case complexity for PLCS.

## 5.3 Practical Considerations

The results in this paper address the worst-case use-case of cluster-sending: the exchange of a *single* value between clusters in complete isolation without *any* knowledge on the likelihood of specific replicas to be faulty. Practical use-cases

typically provide additional knowledge that can be used to further fine-tune the cluster-sending protocols. E.g., if multiple values are to be exchanged in consecutive steps, then one can start the cluster-sending of the *next* value by first attempting to cluster-send via the previously-successful replica pair and by skipping any replica pairs that have failed (in preceding rounds). Likewise, if the likelihood of replicas to be faulty is known to be skewed, then one can incorporate the skew in the *fully at random* selection of replica pairs to maximize the likelihood of selection non-faulty replica pairs.

The cluster-sending problem we consider here only considers sending a value from one cluster to another cluster. The cluster-sending problem can be generalized in two ways toward a multi-cluster-sending protocol:

1. Cluster $C_1$ sends a value $v$ to multiple other clusters $C_{2,1}, \ldots, C_{2,n}$, and the success of sending $v$ to any of the clusters $C_{2,i}$ is independent of the success of sending $v$ to any of the other clusters $C_{2,j}$, $1 \leq i \neq j \leq n$. In this case, the cluster-sending steps to each cluster are independent of each other. Hence, the solutions we present in this paper can be applied in a straightforward manner.

2. Cluster $C_1$ sends a value $v$ to multiple other clusters $C_{2,1}, \ldots, C_{2,n}$ and these clusters only successfully receive $v$ if all clusters do so. This problem is much more akin the traditional *commit problem* as solved by two-phase commit protocols and three-phase commit protocols in the non-resilient setting [14, 31, 36]. One can use cluster-sending as a fundamental *resilient communication primitive* on top of which one can implement resilient versions of these commit protocols (see, e.g., as is the focus of BYSHARD [19, 21]).

## 6  Asynchronous Communication

In the previous sections, we introduced PCS, PPCS, and PLCS, three probabilistic cluster-sending protocols with expected constant message complexity. To simplify presentation, we have presented their design with respect to a synchronous environment. Next, we consider their usage in environments with asynchronous inter-cluster communication due to which messages can get arbitrary delayed, duplicated, or dropped.

We notice that the presented protocols *only* depend on synchronous communication to minimize communication: at the core of the correctness of PCS, PPCS, and PLCS is the cluster-sending step performed by CS-STEP, which does not make any assumptions on communication (Proposition 3.1). Consequently, PCS, PPCS, and PLCS can easily be generalized to operate in environments with asynchronous communication.

First, we observe that message duplication and out-of-order delivery have no impact on the cluster-sending step performed by CS-STEP. Hence, we do not need to take precautions against such asynchronous behavior. Furthermore, if communication is asynchronous, but reliable (messages do not

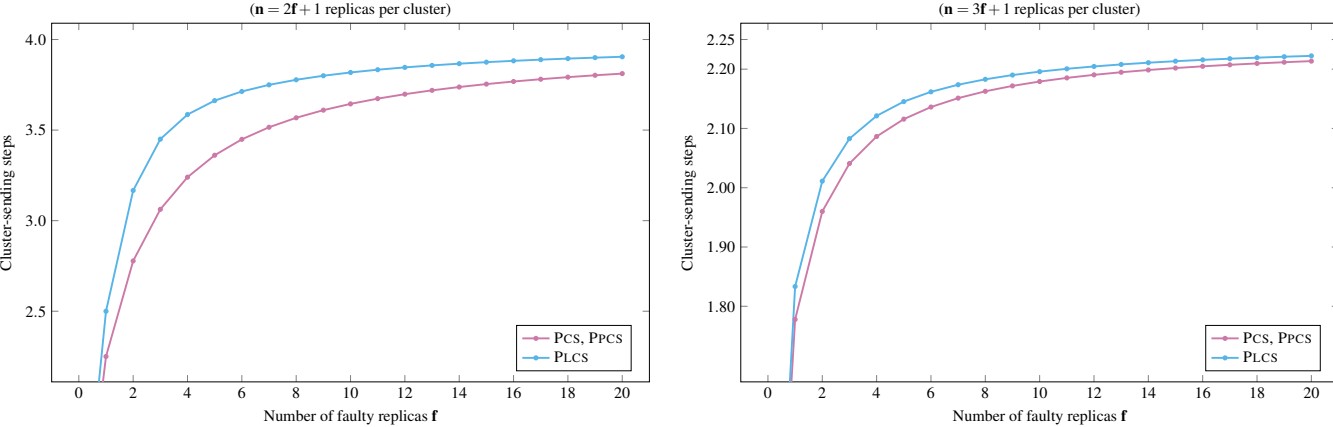

Figure 6: Comparison of the expected-case complexity of PLCS and PCS as a function of the number of faulty replicas.

get lost, but can get duplicated, be delivered out-of-order, or get arbitrarily delayed), both PPCS and PLCS will be able to always perform cluster-sending in a finite number of steps.

If communication is asyncrhonous and unreliable (messages sent between non-faulty replicas can get lost and all cluster-sending steps can fail), then the presented *syncrhonous* protocols can fail and, hence, need to be adjusted to the asyncrhonous environment in which they are deployed. The best way in which a probabilistic cluster-sending solution can deal with *unreliable* asynchronous communication depends on the model of asynchronous communication one is optimizing for.

As an example, we illustrate cluster-sending with a basic model of unreliable communication in which each communication step can independently fail with a probability of $p_{fail}$. In this setting, replicas in $C_1$ can simply continue cluster-sending steps until a step succeeds, which will eventually happen. Similar to the proof of Theorem 4.2, we can prove the following:

**Theorem 6.1.** *Let $C_1, C_2$ be disjoint clusters. If each communication step can independently fail with a probability of $p_{fail}$, then $PCS(C_1, C_2, v)$ is expected to result in cluster-sending $v$ from $C_1$ to $C_2$ in $(\mathbf{n}_{C_1}\mathbf{n}_{C_2})/(\mathbf{nf}_{C_1}\mathbf{nf}_{C_2}) \cdot \frac{1}{(1-p_{fail})^2}$ cluster-sending steps.*

*Sketch.* There are two communication steps between $R_1 \in C_1$ and $R_2 \in C_2$. With a message loss of $p_{fail}$ between the clusters, the probability $q$ that no communication failure happens during a cluster-sending step between two non-faulty replicas is $(1 - p_{fail})^2$. ☐

Not all models of asynchronous communication allow for such a precise characterization of the expected-case complexity for cluster-sending: often, communication steps are not fully independent as the communication model allows for *periods* in which no communication will succeed. This is the case for the *partial synchrony model*, which is often employed by primary-backup consensus protocols such as PBFT. In the partial synchrony model, *unreliable periods of communication* are followed by *sufficiently-long periods of reliable communication*, during which cluster-sending with the presented algorithms will always succeed. During arbitrary-length periods of unreliable of communication, a cluster-sending attempt can perform an arbitrary number of cluster-sending steps.

Although the partial synchrony model and many other asynchronous communication models do not allow for a precise characterization of the expected-case complexity, we can still use our protocols to provide cluster-sending eventually: running the algorithms until successful cluster-sending is performed, by simply continuing until a step succeeds (PCS) or by rerunning the protocol until a step succeeds (PPCS, and PLCS), will assure success as soon as communication becomes reliable.

We note that if communication is asynchronous, then messages can get arbitrarily delayed. Fortunately, practical environments operate with large periods of reliable communication in which the majority of the messages arrive within some bounded delay unknown to $C_1$ and $C_2$. Hence, replicas in $C_1$ can simply assume some delay δ. If this delay is too short, then a cluster-sending step can *appear to fail* simply because the proof of receipt is still under way. In this case, cluster-sending will still be achieved when the proof of receipt arrives, but spurious cluster-sending steps can be initiated in the meantime. To reduce the number of such spurious cluster-sending steps, all non-faulty replicas in $C_1$ can use *exponential backoff* to increase the message delay δ toward some reasonable upper bound (e.g., 100 s).

Finally, asynchronous environments often necessitate rather high assumptions on the message delay δ. Consequently, the duration of a single failed cluster-sending step performed by CS-STEP will be high. Here, a trade-off can be made between *message complexity* and *duration* by starting several rounds of the cluster-sending step at once, which will sharply reduce the duration of the protocol with only a constant increase in expected message complexity. For example,

if PCS is expected to perform *four* cluster-sending steps, then one can reduce the expected-case duration of cluster-sending from the duration of *four cluster-sending steps* to the duration of *one cluster-sending step* by simply performing four cluster-sending steps in parallel (of which one is expected to succeed). In the presence of unreliable communication, one can increase the number of parallel cluster-sending steps in accordance with Theorem 6.1 to keep the duration of cluster-sending low.

## 7 Performance evaluation

In the previous sections, we introduced probabilistic cluster-sending protocols with expected-case constant message complexity. To gain further insight in the performance attainable by these protocols, especially in environments with unreliable communication, we implemented these protocols in a simulated sharded resilient environment that allows us to control the faulty replicas and the message loss rates.[3] As a baseline of comparison, we also evaluated three cluster-sending protocols from the literature:

1. The *worst-case optimal cluster-sending protocol* PBS-CS of Hellings et al. [18, 20] that can perform cluster-sending using only $\mathbf{f}_{C_1} + \mathbf{f}_{C_2} + 1$ messages, which is worst-case optimal. This protocol requires reliable communication.

2. The *broadcast-based cluster-sending protocol* of CHAINSPACE [1] that can perform cluster-sending using $\mathbf{n}_{C_1}\mathbf{n}_{C_2}$ messages. This protocol requires reliable communication.

3. The *global sharing protocol* of GEOBFT [16], an optimistic cluster-sending protocol that assumes that each cluster uses a primary-backup consensus protocol (e.g., PBFT [6]) and optimizes for the case in which the coordinating primary of $C_1$ is non-faulty. In this optimistic case, GEOBFT can perform cluster-sending using only $\mathbf{f}_{C_2} + 1$ messages. To deal with faulty primaries and unreliable communication, GEOBFT employs a costly remote view-change protocol, however.

We refer to Figure 2 for an analytical comparison between these three cluster-sending protocols and our three probabilistic cluster-sending protocols.

In each experiment, we measured the number of messages exchanged in 10000 runs of the cluster-sending protocol under consideration. In specific, in each run we measure the number of messages exchanged when sending a value *v* from a cluster $C_1$ to a cluster $C_2$ with $\mathbf{n}_{C_1} = \mathbf{n}_{C_2} = 3\mathbf{f}_{C_1} + 1 = 3\mathbf{f}_{C_2} + 1$, and we aggregate this data over 10000 runs. The messages exchanged is an objective measure of the performance of the

cluster-sending protocols under consideration that is independent of the environment (e.g., network bandwidth, message delays) and the application use-case for which cluster-sending is used. As we use equal-sized clusters, we have $\Phi_{\min}(C_1, C_2) = \Phi_{\max}(C_1, C_2)$ and, hence, we use a singe instance of PLCS.

Next, we detail the two experiments we performed and look at their results.

### 7.1 Performance of Cluster-Sending Protocols

In our first experiment, we measure the number of messages exchanged as a function of the number of faulty replicas. In this case, we assumed reliable communication, due to which we could include all six protocols. The results of this experiment can be found in Figure 7.

As is clear from the results, our probabilistic cluster-sending protocols are able to perform cluster-sending with only a constant number of messages exchanged. Furthermore, we see that the performance of our cluster-sending protocols matches the theoretical expected-case analysis in this paper and closely follows the expected performance illustrated in Figure 6 (note that Figure 6 plots cluster-sending steps and each cluster-sending step involves the exchange of *two* messages between clusters).

As all other cluster-sending protocols have a linear (PBS-CS and GEOBFT) or quadratic (CHAINSPACE) message complexity, our probabilistic cluster-sending protocols outperform the other cluster-sending protocols. This is especially the case when dealing with bigger clusters, in which case the expected-case constant message complexity of our probabilistic cluster-sending protocols shows the biggest advantage. Only in the case of the smallest clusters can the other cluster-sending protocols outperform our probabilistic cluster-sending protocols, as PBS-CS, GEOBFT, and CHAINSPACE use reliable communication to their advantage to eliminate any acknowledgment messages send from the receiving cluster to the sending cluster. We believe that the slightly higher cost of our probabilistic cluster-sending protocols in these cases is justified, as our protocols can effectively deal with unreliable communication.

### 7.2 Message Loss

In our second experiment, we measure the number of messages exchanged as a function of the number of faulty replicas and as a function of the message loss (in percent) *between the two clusters*. We only focus on message loss between clusters, and we assume that consensus steps *within a cluster* always succeed. In this case, we only included our probabilistic cluster-sending protocols, as PBS-CS and CHAINSPACE both assume reliable communication and GEOBFT is only able to perform recovery via remote view-changes in periods

---

[3]The full implementation of this experiment is available at https://www.jhellings.nl/projects/csp/.

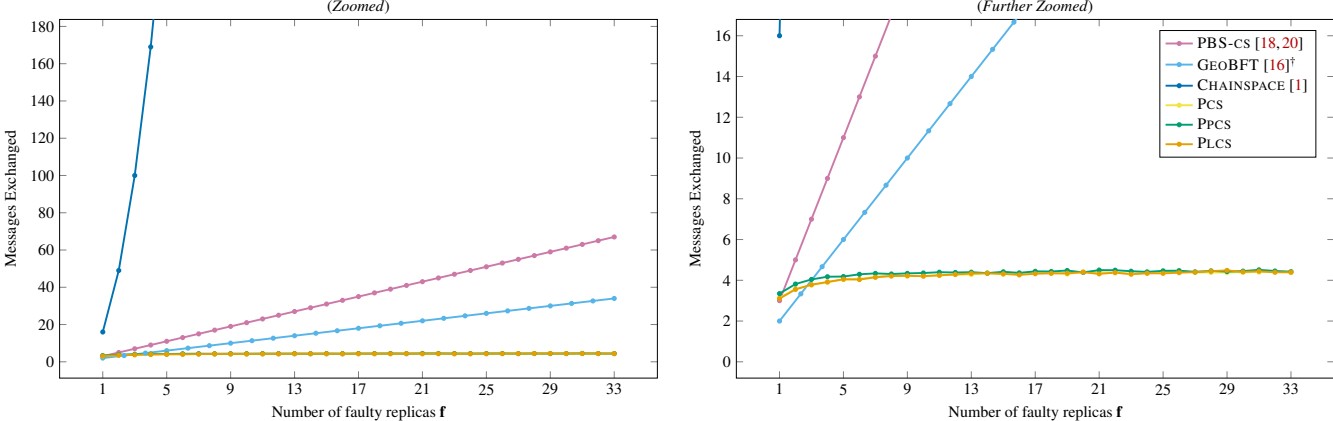

Figure 7: A comparison of the number of message exchange steps as a function of the number of faulty replicas in both clusters by our probabilistic cluster-sending protocols PCS, PPCS, and PLCS, and by three protocols from the literature. For each protocol, we measured the number of message exchange steps to send a value between two equally-sized clusters (average of 10 000 runs), each cluster having $n = 3\mathbf{f} + 1$ replicas. †The results for GEOBFT are a plot of the best-case optimistic phase of that protocol.

of reliable communication. The results of this experiment can be found in Figure 8.

Theorem 6.1 can be used to derive the expected-case complexity of PCS in each of the cases. For example, with a message loss of $p_{\text{fail}} = 30\%$, the probability $q$ that no communication failure happens during a cluster-sending step is $q = 0.49$. Hence, when compared to the case without message loss, we expect an increase in the message complexity by a factor $1/q \approx 2$. Our measurements on PCS show that this expected increase also happens in practice. For PPCS and PLCS, we did not provide a theoretical analysis of the expected-case complexity. Still, in practice we see a similar behavior as with PCS: we see an increase in the message complexity by a factor $1/q \approx 2$ for all protocols.

This observation extends to other probabilities of message loss. Although the probabilistic arguments underpinning the expected-case cost of, on the one hand, PCS and PPCS and, on the other hand, PLCS are vastly different, the results of these experiments show that across the board, the practical performance of the three protocols is similar.

These results further underline the practical benefits of each of the probabilistic cluster-sending protocols, especially for larger clusters: even in the case of high message loss rates, each of our probabilistic cluster-sending protocols are able to outperform the cluster-sending protocols PBS-CS, CHAINSPACE, and GEOBFT, which can only operate with reliable-communication.

## 8   Related Work

Although there is abundant literature on distributed systems and on consensus-based resilient systems (e.g., [2, 5, 8, 15, 17, 31, 37]), there is only limited work on communication *between* resilient systems [1, 16, 18, 20]. In the previous sec-

tion, we have already compared PCS, PPCS, and PLCS with the worst-case optimal cluster-sending protocols of Hellings et al. [18, 20], the optimistic cluster-sending protocol of GEOBFT [16], and the broadcast-based cluster-sending protocols of CHAINSPACE [1]. Furthermore, we notice that *cluster-sending* can be solved using well-known Byzantine primitives such as consensus, interactive consistency, and Byzantine broadcasts [6, 9, 27]. These primitives are much more costly than cluster-sending protocols, however, and require huge amounts of communication between all involved replicas.

In parallel to the development of traditional resilient systems and permissioned blockchains, there has been promising work on sharding in permissionless blockchains such as BITCOIN [28] and ETHEREUM [38]. Examples include techniques for enabling reliable cross-chain coordination via sidechains, blockchain relays, atomic swaps, atomic commitment, and cross-chain deals [12, 13, 22, 24, 25, 39, 40]. Unfortunately, these techniques are deeply intertwined with the design goals of permissionless blockchains in mind (e.g., cryptocurrency-oriented), and are not readily applicable to traditional consensus-based Byzantine clusters.

## 9   Conclusion

In this paper, we presented probabilistic cluster-sending protocols that each provide highly-efficient solutions to the cluster-sending problem. Our probabilistic cluster-sending protocols can facilitate communication between Byzantine fault-tolerant clusters with expected constant communication between clusters. For practical environments, our protocols can support worst-case linear communication between clusters, which is optimal, and deal with asynchronous and unreliable communication. The low practical cost of our cluster-

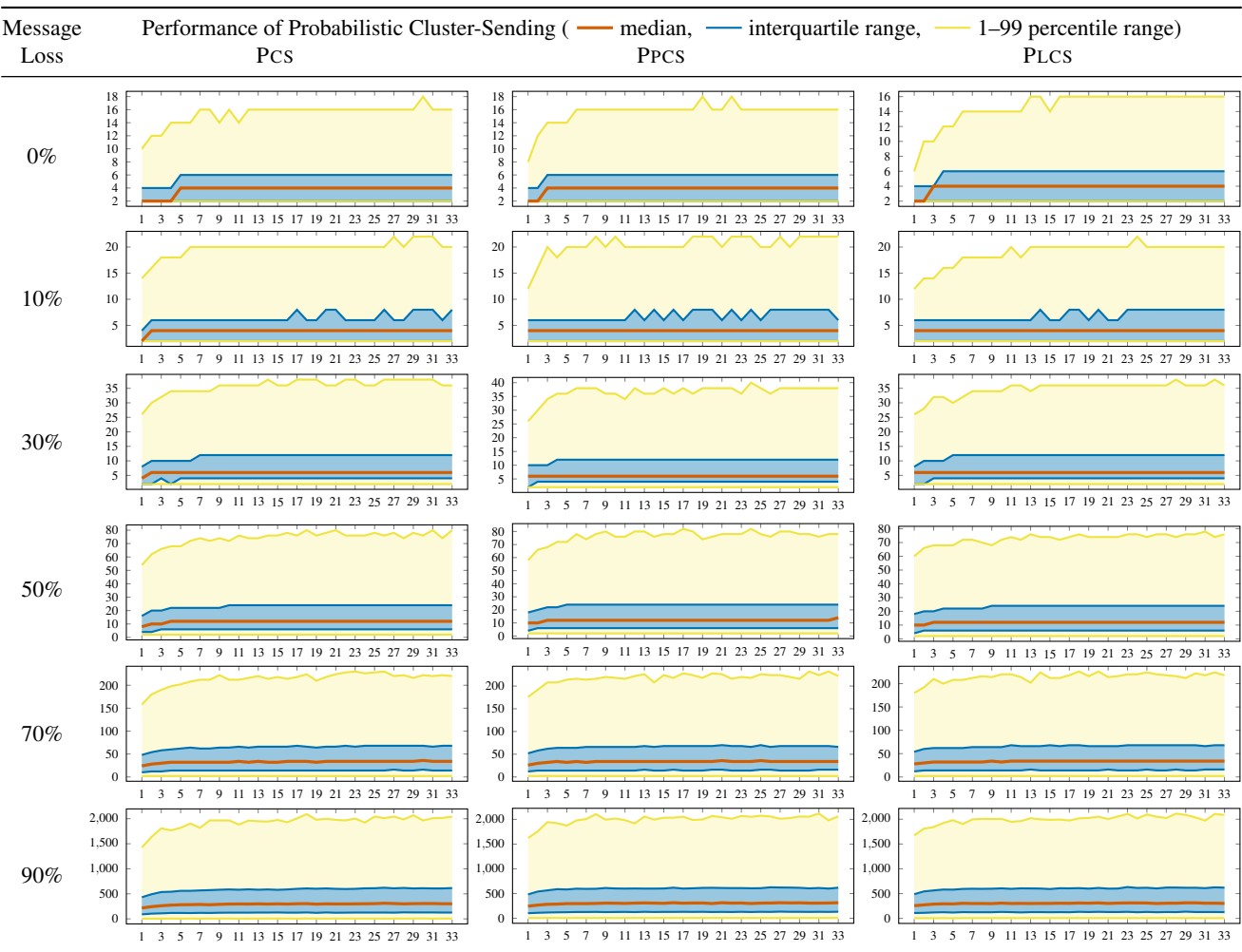

Figure 8: A comparison of the number of message exchange steps as a function of the number of faulty replicas in both clusters and of the message loss by our probabilistic cluster-sending protocols Pcs, Ppcs, and Plcs. For each protocol, we measured the number of message exchange steps to send 10 000 values between two equally-sized clusters, each cluster having $n = 3\mathbf{f} + 1$ replicas, after which we aggregated the measurements to obtain a summary of the distribution of messages exchanged.

sending protocols further enables the development and deployment of high-performance systems that are constructed out of Byzantine fault-tolerant clusters, e.g., fault-resilient geo-aware sharded data processing systems.

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

# A  The proof of Lemma 5.2

To get the intuition behind the closed form of Lemma 5.2, we take a quick look at the combinatorics of *list-merging*. Notice that we can merge lists $S$ and $T$ together by either first taking an element from $S$ or first taking an element from $T$. This approach towards list-merging yields the following recursive solution to the list-merge problem:

$$\mathbb{M}(v, w) = \begin{cases} \mathbb{M}(v-1, w) + \mathbb{M}(v, w-1) & \text{if } v > 0 \text{ and } w > 0; \\ 1 & \text{if } v = 0 \text{ or } w = 0. \end{cases}$$

Consider lists $S$ and $T$ with $|S| = v$ and $|T| = w$ distinct values. We have $|\text{perms}(S)| = v!$, $|\text{perms}(T)| = w!$, and $|\text{perms}(S \cup T)| = (v+w)!$. We observe that every list-merge of $(P_S, P_T) \in \text{perms}(S) \times \text{perms}(T)$ is a unique value in $\text{perms}(S \cup T)$. Furthermore, every value in $\text{perms}(S \cup T)$ can be constructed by such a list-merge. As we have $|\text{perms}(S) \times \text{perms}(T)| = v!w!$, we derive the closed form

$$\mathbb{M}(v, w) = \frac{(v+w)!}{(v!w!)}$$

of Lemma 5.2. Next, we formally prove this closed form.

*Proof.* We prove this by induction. First, the base cases $\mathbb{M}(0, w)$ and $\mathbb{M}(v, 0)$. We have

$$\mathbb{M}(0, w) = \frac{(0+w)!}{0!w!} = \frac{w!}{w!} = 1;$$
$$\mathbb{M}(v, 0) = \frac{(v+0)!}{v!0!} = \frac{v!}{v!} = 1.$$

Next, we assume that the statement of the lemma holds for all non-negative integers $v', w'$ with $0 \le v' + w' \le j$. Now consider non-negative integers $v, w$ with $v + w = j + 1$. We assume that $v > 0$ and $w > 0$, as otherwise one of the base cases applies. Hence, we have

$$\mathbb{M}(v, w) = \mathbb{M}(v-1, w) + \mathbb{M}(v, w-1).$$

We apply the induction hypothesis on the terms $\mathbb{M}(v-1, w)$ and $\mathbb{M}(v, w-1)$ and obtain

$$\mathbb{M}(v, w) = \left( \frac{((v-1)+w)!}{(v-1)!w!} \right) + \left( \frac{(v+(w-1))!}{v!(w-1)!} \right).$$

Next, we apply $x! = x(x-1)!$ and simplify the result to obtain

$$
\begin{aligned}
\mathbb{M}(v,w) &= \left(\frac{v(v+w-1)!}{v!w!}\right) + \left(\frac{w(v+w-1)!}{v!w!}\right) \\
&= \left(\frac{(v+w)(v+w-1)!}{v!w!}\right) = \frac{(v+w)!}{v!w!},
\end{aligned}
$$

which completes the proof. □

## B The simplification of $\mathbb{F}(n,m_1,m_2,k)$

Let $g$ be the expression

$$
\begin{aligned}
&b_1!^2\binom{m_1}{b_1}\binom{n-m_2}{b_1}b_2!^2\binom{n-m_1}{b_2}\binom{m_2}{b_2}. \\
&\mathbb{M}(b_1,b_2)b_{1,2}!^2\mathbb{M}(b_1+b_2,b_{1,2})(n-k)!^2\mathbb{M}(k,n-k),
\end{aligned}
$$

as stated right above Lemma 5.3. We will show that $g$ is equivalent to the closed form of $\mathbb{F}(n,m_1,m_2,k)$, as stated in Lemma 5.3.

*Proof.* We use the shorthands $\mathbf{T}_1 = \binom{m_1}{b_1}\binom{n-m_2}{b_1}$ and $\mathbf{T}_2 = \binom{n-m_1}{b_2}\binom{m_2}{b_2}$, and we have

$$
\begin{aligned}
g = b_1!^2\mathbf{T}_1 b_2!^2\mathbf{T}_2 \cdot \\
\mathbb{M}(b_1,b_2)b_{1,2}!^2\mathbb{M}(b_1+b_2,b_{1,2})(n-k)!^2\mathbb{M}(k,n-k).
\end{aligned}
$$

We apply Lemma 5.2 on terms $\mathbb{M}(b_1,b_2)$, $\mathbb{M}(b_1+b_2,b_{1,2})$, and $\mathbb{M}(k,n-k)$, apply $k = b_1+b_2+b_{1,2}$, and simplify to derive

$$
\begin{aligned}
g = b_1!^2\mathbf{T}_1 b_2!^2\mathbf{T}_2 \cdot \\
\frac{(b_1+b_2)!}{b_1!b_2!}b_{1,2}!^2\frac{(b_1+b_2+b_{1,2})!}{(b_1+b_2)!b_{1,2}!}(n-k)!^2\frac{(k+n-k)!}{k!(n-k)!} \\
= b_1!\mathbf{T}_1 b_2!\mathbf{T}_2 b_{1,2}!(n-k)!n!.
\end{aligned}
$$

Finally, we expand the binomial terms $\mathbf{T}_1$ and $\mathbf{T}_2$, apply $b_{1,2} = m_1-b_1 = m_2-b_2$ and $k = m_1+b_2 = m_2+b_1$, and simplify to derive

$$
\begin{aligned}
g &= b_1!\frac{m_1!}{b_1!(m_1-b_1)!}\frac{(n-m_2)!}{b_1!(n-m_2-b_1)!} \cdot \\
&\quad b_2!\frac{(n-m_1)!}{b_2!(n-m_1-b_2)!}\frac{m_2!}{b_2!(m_2-b_2)!} \cdot \\
&\quad b_{1,2}!(n-k)!n! \\
&= \frac{m_1!}{b_{1,2}!}\frac{(n-m_2)!}{b_1!(n-k)!}\frac{(n-m_1)!}{b_2!(n-k)!}\frac{m_2!}{b_{1,2}!}b_{1,2}!(n-k)!n! \\
&= \frac{m_1!m_2!(n-m_1)!(n-m_2)!n!}{b_1!b_2!b_{1,2}!(n-k)!},
\end{aligned}
$$

which completes the proof. □

## C The Closed Form of $\mathbb{E}(2f+1,f,f)$

Here, we shall prove that

$$
\mathbb{E}(2f+1,f,f) = 4 - \frac{2}{(f+1)} - \frac{f!^2}{(2f)!}.
$$

*Proof.* By Proposition 5.2 and some simplifications, we have

$$
\begin{aligned}
\mathbb{E}(2f+1,f,f) &= \frac{1}{(2f+1)!^2} \cdot \\
&\left(\sum_{k=f}^{2f}\frac{2f+1}{2f+1-k}\frac{f!^2(f+1)!^2(2f+1)!}{(k-f)!^2(2f-k)!(2f+1-k)!}\right).
\end{aligned}
$$

First, we apply $x! = x(x-1)!$, simplify, and obtain

$$
\begin{aligned}
\mathbb{E}(2f+1,f,f) &= \frac{f!^2(2f+1)}{(2f+1)!} \cdot \\
&\left(\sum_{k=f}^{2f}\frac{(f+1)!^2}{(k-f)!^2(2f+1-k)!^2}\right) \\
&= \frac{f!^2}{(2f)!}\left(\sum_{k=0}^{f}\frac{(f+1)!^2}{k!^2(f+1-k)!^2}\right) \\
&= \frac{f!^2}{(2f)!}\left(\sum_{k=0}^{f}\binom{f+1}{k}^2\right).
\end{aligned}
$$

Next, we apply $\binom{m}{n} = \binom{m}{m-n}$, extend the sum by one term, and obtain

$$
\begin{aligned}
\mathbb{E}(2f+1,f,f) &= \frac{f!^2}{(2f)!} \cdot \\
&\left(\left(\sum_{k=0}^{f+1}\binom{f+1}{k}\binom{f+1}{f+1-k}\right) - \binom{f+1}{f+1}\binom{f+1}{0}\right).
\end{aligned}
$$

Then, we apply Vandermonde's Identity to eliminate the sum and obtain

$$
\mathbb{E}(2f+1,f,f) = \frac{f!^2}{(2f)!}\left(\binom{2f+2}{f+1} - 1\right).
$$

Finally, we apply straightforward simplifications and obtain

$$
\begin{aligned}
\mathbb{E}(2f+1,f,f) &= \frac{f!^2}{(2f)!} \frac{(2f+2)!}{(f+1)!(f+1)!} - \frac{f!^2}{(2f)!} \\
&= \frac{f!^2}{(2f)!} \frac{(2f)!(2f+1)(2f+2)}{f!^2(f+1)^2} - \frac{f!^2}{(2f)!} \\
&= \frac{(2f+1)(2f+2)}{(f+1)^2} - \frac{f!^2}{(2f)!} \\
&= \frac{(2f+2)^2}{(f+1)^2} - \frac{2f+2}{(f+1)^2} - \frac{f!^2}{(2f)!} \\
&= \frac{4(f+1)^2}{(f+1)^2} - \frac{2(f+1)}{(f+1)^2} - \frac{f!^2}{(2f)!} \\
&= 4 - \frac{2}{f+1} - \frac{f!^2}{(2f)!},
\end{aligned}
$$

which completes the proof. $\qquad\square$

