# OpenReview forum: "Solution: Byzantine Cluster-Sending in Expected Constant Cost and Constant Time"
_JSYS/2023/March_Papers — Accept (with shepherding)_

### Official Review · Reviewer_tpVP · 2023-04-11

**Decision:**

Strong accept: excellent paper that will help the community

**Strengths:**

The main contribution of this paper are:
* Give a protocol with an expected constant inter-cluster messages.
* Give a protocol with an expected constant and worst-case linear inter-cluster messages.
* Brings attention to the cluster sending problem when local and global communication have huge differences in costs.


**Weaknesses:**

* nothing I can think of for now.

**Detailed Comments:**

Small Typo:

* There seem to be a missing factorial on page 18 (x=x(x-1)!)


**Expertise:**

Published in this area in the last 5 years

**Summary Of Review:**

This paper studies the cluster-sending problem defined from Hellings et al. [DISC 2019].
The authors addressed the previous review comments. The model description, the discussions in asynchronous settings, the discussion for multi-cluster cluster-sending problem, and the figures are improved. I recommend accepting this paper.



**Useful:**

yes

---

### Official Review · Reviewer_toSZ · 2023-04-11

**Decision:**

Strong accept: excellent paper that will help the community

**Strengths:**

It substantially improves the message complexity.

**Weaknesses:**

Does not provide any result on bit complexity, i.e., total number of bits transmitted.

**Detailed Comments:**

Bit complexity is also an improvant complexity measure as it has to do with the throughput of the systems. Some discussion on bit complexity should be added.

**Expertise:**

Published in this area in the last 5 years

**Summary Of Review:**

The authors provide a new algorithm, "probabilistic cluster-sending techniques," to improve the message complexity from previously linear in the size of the cluster involved to constant. For practical systems, this is a great improvement. I am happy to see that the authors addressed my previous comments.

**Useful:**

yes

---

### Official Review · Reviewer_v81j · 2023-04-12

**Decision:**

Strong accept: excellent paper that will help the community

**Strengths:**

Improves the expected message complexity of a relevant problem

Can be applied to actual systems

Includes theory and experiments


**Weaknesses:**

Nothing to point out

**Detailed Comments:**

Nothing to point out

**Expertise:**

Follow the literature closely, last published 5+ years ago

**Summary Of Review:**

The paper deals with the practical need to support the sharding of data among independent fault-tolerant clusters. It improves the message complexity of byzantine fault tolerant sending of values among clusters.

The authors addressed the issues raised in prior reviews.

**Useful:**

yes

---

### Official Review · Reviewer_rYNz · 2023-04-14

**Decision:**

Strong accept: excellent paper that will help the community

**Strengths:**

The solution is simple.
The paper presents evaluation results.

**Weaknesses:**

The discussion on asynchrony is hand-wavy.

**Detailed Comments:**

This revision has addressed previous reviewer concerns.

**Expertise:**

Actively publishing in this area

**Summary Of Review:**

This paper proposes a simple solution to the cluster sending problem.

**Useful:**

yes

---

### Meta-Review · Area_Chair_13Wj · 2023-04-17

**Recommendation:** Accept
**Confidence:** 5

**Metareview:**

Dear Authors,

Thank you again for submitting the revised version of your manuscript titled "Byzantine Cluster-Sending in Expected Constant Cost and Constant Time." Reviewers have agreed to accept your paper. Congratulations! Thank you for the hard work you have put on producing this version.

As per JSys rules, your paper will undergo shepherding to produce the final camera ready version. Although all reviewers showed excitement in accepting the paper, there are still two suggestions we would recommend to address. One reviewer noticed a missing factorial on page 18 (x=x(x-1)!). Another reviewer inquired about the possibility of discussing the bit complexity of your proposal.

The camera ready version is due in one month, May 17, 2023. To allow timely shepherding, please submit the revised version, with annotated changes, one week before, May 10, 2023, or earlier of course.

If you have any questions or comments, please do not hesitate to reach out to us. Thank you again for submitting to JSys, and congratulations!

Lewis Tseng and Roberto Palmieri
Area chairs of JSys

---

### Decision · Program_Chairs · 2023-04-27

**Decision:**

Accept (with shepherding)

**Comment:**

Congratulations on getting your manuscript accepted!

The meta-review contains details about the reviewers' expectations for the final version. Please reach out to the Area Chairs if anything is unclear.

Romain Jacob, JSys Editor-in-Chief